# Gradient-based Sample Selection for Faster Bayesian Optimization

## Abstract

Bayesian optimization (BO) is an effective technique for black-box optimization. However, its applicability is typically limited to moderate-budget problems due to the cubic complexity of fitting the Gaussian process (GP) surrogate model. In large-budget scenarios, directly employing the standard GP model faces significant challenges in computational time and resource requirements. In this paper, we propose Gradient-based Sample Selection Bayesian Optimization (GSSBO), a subset-maintenance approach designed to enhance the computational efficiency of BO. Here, "gradient-based" refers to response-gradient sensitivity embeddings induced by the GP log marginal likelihood, not derivative observations of the objective function. The GP model is constructed on a selected set of samples instead of the whole dataset. These samples are selected using GP-specific sensitivity embeddings and a cosine-diversity rule to encourage embedding-space diversity and reduce redundancy in the retained subset. We provide a conditional theoretical analysis of the subset-fitted GP surrogate induced by a fixed retained subset and derive posterior-approximation bounds in terms of subset-dependent residual quantities. Experiments on synthetic and real-world tasks show empirical reductions in GP-fitting cost while achieving competitive optimization performance in the reported settings.

## 1 Introduction

Bayesian optimization (BO) (Frazier, 2018; Couckuyt et al., 2022) is a successful approach to black-box optimization that has been applied in a wide range of applications, such as hyperparameter optimization (Luo et al., 2021; 2024) and resource exploration (Branke et al., 2024). BO's strength lies in its ability to represent the unknown objective function through a surrogate model and by optimizing an acquisition function (Garnett, 2023; Wang et al., 2023; Zhang et al., 2026). BO consists of a surrogate model, which provides a global prediction for the unknown objective function, and an acquisition function that serves as a criterion to determine the next sample to evaluate. In particular, the Gaussian process (GP) model is often preferred as the surrogate model due to its versatility and reliable uncertainty estimation. However, the GP model often suffers from large data sets, making it more suitable for small-budget scenarios (Binois and Wycoff, 2022). To fit a GP model, the dominant cost comes from factorizing the GP observation covariance matrix and solving the associated linear systems, which scales as $\mathcal{O}(n^3)$, where $n$ is the number of data samples. As the sample set grows, the computational burden increases substantially. This limitation poses a significant challenge for scaling BO to real-world problems with large sample sets.

Despite the various approaches to improve the computational efficiency of BO, including parallel BO (González et al., 2016; Daulton et al., 2021; 2020; Eriksson et al., 2019), kernel or local approximation methods (Kim et al., 2021; Jimenez and Katzfuss, 2023; Williams and Seeger, 2000), and sparse or variational GP surrogates (Lawrence et al., 2002; Hensman et al., 2013; Leibfried et al., 2020; McIntire et al., 2016), the computational overhead of repeated GP updates remains a practical bottleneck in large-budget BO (Shahriari et al., 2015). Existing scalable BO approaches typically reduce this cost either by altering the covariance representation or by constructing an alternative sparse posterior, whereas our method instead maintains an explicit retained subset of observations and refits the GP directly on that subset.

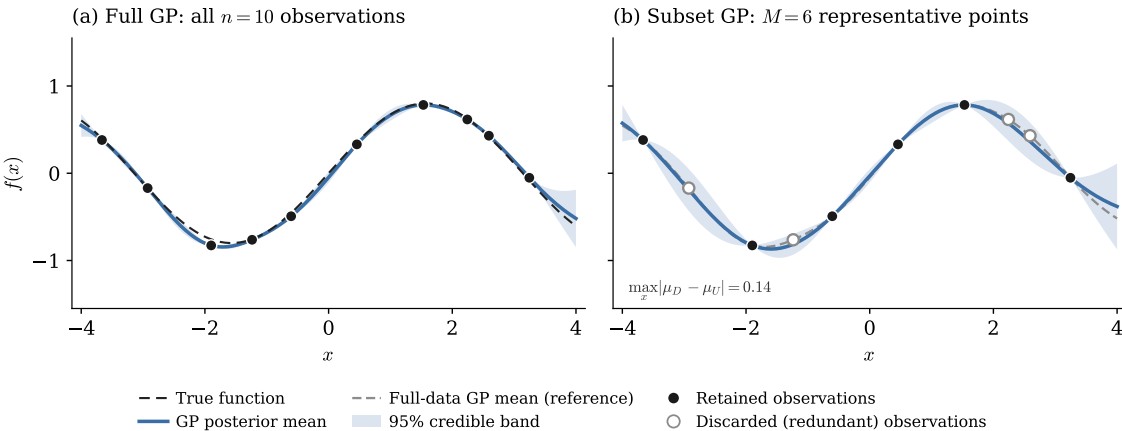

Figure 1: A representative subset preserves the GP posterior at lower fitting cost. (a) GP fitted on all $n = 10$ observations. (b) GP fitted on a representative subset of $M = 6$ observations under the same kernel hyperparameters; the discarded observations (hollow markers) are redundant given nearby retained ones. The subset posterior stays close to the full-data posterior, illustrating that pruning redundant observations changes the posterior little while reducing the cost of GP refitting.

During the iterative search process of BO, nearby or redundant observations can become less useful for subsequent model updates as the dataset grows. This motivates subset-maintenance strategies that retain a compact set of observations for GP refitting. Figure 1 gives a toy illustration in which a smaller selected subset can capture the main posterior trend, while the quality of this approximation depends on the retained subset. In this paper, we study a subset-maintenance strategy for BO in which a fixed-size buffer of observations is retained and the GP surrogate is refit only on that subset. Inspired by replay-buffer maintenance ideas from continual learning (Aljundi et al., 2019), we use GP-specific sensitivity embeddings as a heuristic for encouraging diversity and reducing redundancy in the retained buffer. This retained subset is then used to fit the GP model, reducing the cost of GP updates while aiming to retain competitive optimization performance. To the best of our knowledge, this is among the first BO methods to use sensitivity embeddings to maintain a compact observation subset through a cosine-diversity rule and refit the GP only on that subset. We summarize our main contributions as follows:

- **Efficient computations.** We propose Gradient-based Sample Selection Bayesian Optimization (GSSBO), which maintains a fixed-size buffer of observed samples using sensitivity embeddings and a cosine-diversity rule, and refits the GP surrogate only on that subset, thereby reducing the cost of GP updates in large-budget scenarios.

- **Theoretical analysis.** We analyze how subset fitting with a fixed retained subset affects the posterior mean and uncertainty of a GP, and derive approximation-error bounds in terms of subset-dependent residual quantities. These results are conditional diagnostics that hold for any fixed retained subset under shared hyperparameters; they do not prove that the cosine-diversity selection rule controls the residual quantities appearing in the bounds.

- **Empirical evaluation.** We conduct numerical experiments on synthetic and real-world test problems. These results suggest that the proposed subset-maintenance heuristic can provide a favorable empirical trade-off between BO performance and GP-fitting cost.

## 2 Related Works

**BO with Resource Challenges.** In practical applications, BO faces numerous challenges, including high evaluation costs, input-switching costs, resource constraints, and high-dimensional search spaces. Researchers have proposed a variety of methods to address these issues. For instance, parallel BO employs batch sampling

to improve efficiency in large-scale or highly concurrent scenarios (González et al., 2016; Daulton et al., 2021; 2020; Eriksson et al., 2019). Kernel approximation methods, such as random Fourier features, map kernels onto lower-dimensional feature spaces, thus accelerating kernel-based approaches (Rahimi and Recht, 2007; Kim et al., 2021). Multi-fidelity BO leverages coarse simulations with a limited number of high-fidelity evaluations to reduce the overall cost (Kandasamy et al., 2016). For high-dimensional tasks, random embeddings or active subspaces help reduce the search dimensionality (Wang et al., 2016; Nayebi et al., 2019). Meanwhile, sparse GP methods reduce computational complexity through inducing points and variational approximations (Lawrence et al., 2002; Hensman et al., 2013; Leibfried et al., 2020; McIntire et al., 2016; Moss et al., 2023). Canonical variational formulations of sparse GP regression were developed by Titsias (2009), and later work further clarified the relationship between inducing-point variational GPs and Nyström approximations (Wild et al., 2021). Our method is different in that it retains an explicit subset of observations and refits the GP on that subset, rather than constructing an inducing-point posterior approximation. However, these approaches can introduce approximation trade-offs or additional implementation complexity in practical BO settings. Calandriello et al. (2022) scale GP optimization by repeatedly evaluating each selected point until its posterior uncertainty falls below a preset threshold, thus limiting the number of active datasets. However, the dataset still grows with time, and the algorithm's dependence on its initial sample set means that low-value points selected early on remain permanently in the model, potentially inflating computational overhead. Compared with inducing-point approximations, retaining an explicit subset of observations keeps the surrogate an exact GP on the retained data: no variational family or evidence-lower-bound approximation is introduced, the posterior form is unchanged, and any acquisition function can be used unmodified. The subset update is a discrete selection step rather than a continuous optimization of inducing locations, avoiding the per-iteration gradient training of variational sparse GPs, and the retained points are actual evaluated observations, which keeps the surrogate interpretable and allows the retained set to be reused directly.

**BO with Gradient Information.** The availability of derivative information can significantly simplify optimization problems. Ahmed et al. (2016) highlight the potential of incorporating gradient information into BO methods and advocate for its integration into optimization frameworks. Wu and Frazier (2016) introduced the parallel knowledge gradient method for batch BO, achieving faster convergence to global optima. They later introduced d-KG (Wu et al., 2017), a new acquisition function that incorporates derivative observations into BO; in that setting, derivative observations can improve the information value under the assumptions studied there. Rana et al. (2017) incorporated GP priors to enable gradient-based local optimization. Chen et al. (2018) proposed a unified particle-optimization framework using Wasserstein gradient flows for scalable Bayesian sampling. Bilal et al. (2020) demonstrated that BO with gradient-boosted regression trees performed well in cloud configuration tasks. Tamiya and Yamasaki (2022) developed stochastic gradient line BO (SGLBO) for noise-robust quantum circuit optimization. Penubothula et al. (2021) found local critical points by querying where the predicted gradient is zero. Zhang and Rodgers (2024) introduced BO of gradient trajectory (BOGAT) for efficient imaging optimization. Makrygiorgos et al. (2025) integrated exact gradient observations into the Bayesian neural network surrogate's training loss. Although these methods leverage gradient information to improve optimization efficiency and performance, they mainly focus on refining the surrogate model or acquisition rule. In contrast, GSSBO uses sensitivity embeddings only as a retained-subset maintenance heuristic; it does not require derivative observations and keeps the BO acquisition unchanged.

**Subset Selection.** Subset selection is a key task in fields such as regression, classification, and model selection, aiming to improve efficiency by selecting a subset of features or data. Random subset selection, a simple and widely used method, involves randomly sampling data, often for cross-validation or bootstrap (Hastie, 2009). Importance-based selection focuses on high-value data points, while active learning targets samples that are expected to provide the most information, improving model learning (Quinlan, 1986). Filter methods rank features using statistical measures such as correlation or variance, selecting the top-ranked ones for modeling (Guyon and Elisseeff, 2003). Narendra and Fukunaga (1977) introduced a branch-and-bound algorithm for efficient feature selection. Yang et al. (2022) proposed dataset pruning, an optimization-based sample selection method that identifies the smallest subset of training data to reduce training costs. Ash et al. (2019) employ the k-means++ algorithm in the gradient space for diversity sampling in active learning.

---

Empirically, the SVGP-based baselines attain higher final cumulative regret than GSSBO on Powell50 in Table 1 ($7.35 \times 10^7$ for SparseGP and $6.36 \times 10^7$ for IPA versus $5.48 \times 10^7$), while costing 65–130% of full-GP runtime versus GSSBO's ∼9% (Table 7).

Oglic and Gärtner (2017) first map each data point into the reproducing kernel Hilbert space (RKHS), then use a max–min coverage strategy in the RKHS to sequentially sample $K$ landmarks, which are employed to construct the Nyström low-rank approximation. This method is dependent on the quality of the selected landmarks. Hayakawa et al. (2023) provide tighter expected error bounds under a continuous measure for the same underlying idea. However, computing the Mercer decomposition in high dimensions incurs substantial computational cost and suffers from severe error degradation.

Our construction is related to gradient-space subset selection, but it uses a different representation. Methods such as Aljundi et al. (2019) and Ash et al. (2019) select data using gradient representations in model or loss space. In GSSBO, the vector feature is instead a column of the GP response-score Jacobian, equivalently a pool-specific observation-precision column. It also differs from RKHS landmark selection such as Oglic and Gärtner (2017) and Hayakawa et al. (2023): the precision-column feature depends on the current observation pool and fitted GP covariance rather than being a fixed feature map of the original kernel. We therefore interpret the vector rule as a greedy diversity heuristic in a precision-derived feature geometry. Because both of these methods perform their approximations in RKHS at high computational expense, they are most suitable for offline, batch-mode resampling. Zhu (2016) proposed a "pilot estimate" to approximate the gradient of the objective function. The core idea is to compute the gradient information corresponding to each data point based on an initial parameter estimate and identify data points with larger gradient values as more "important" samples for subsequent optimization. Despite these advancements, directly applying subset selection methods to BO often yields suboptimal results, necessitating further exploration to integrate subsampling effectively into BO.

## 3 Preliminaries

### 3.1 Bayesian Optimization and Gaussian Processes

BO aims to find the global optimum $x^* \in \mathcal{X}$ of an unknown reward function $f : \mathcal{X} \to \mathbb{R}$ over the $d$-dimensional input space $\mathcal{X} = [0,1]^d$. Throughout this paper, we consider maximization problems, i.e., we seek $x^* = \arg\max_{x \in \mathcal{X}} f(x)$ as quickly as possible. GPs are one of the fundamental components in BO, providing a theoretical framework for modeling and prediction in a black-box function. In each round, a sample $x_t$ is selected based on the current GP's posterior and acquisition function. The observation $(x_t, y_t)$ is then appended to the accumulated dataset $\mathcal{D}$, and the GP surrogate is updated according to these samples. This iterative process of sampling and updating continues until the optimization objectives are achieved or the available budget is exhausted. The key advantage of GPs lies in their nonparametric nature, allowing them to model complex functions without assuming a specific form. GPs are widely used for regression (Gaussian Process Regression (Schulz et al., 2018), GPR) and classification tasks due to their flexibility and ability to provide uncertainty estimates. Formally, a GP can be defined as: $f(x) \sim \mathcal{GP}(\mu(x), k(x, x'))$, where $\mu(x)$ is the mean function, often assumed to be zero, and $k(x, x')$ is the covariance function, defining the similarity between points $x$ and $x'$. It should be noted that the algorithmic complexity of GP updates is $\mathcal{O}(n^3)$, where $n$ is the number of observed samples. As the sample set grows, the computational resources required for these updates can become prohibitively expensive, especially in large-scale optimization problems. A summary of GP regression and the BO loop in our notation, including standard acquisition functions, is provided in Appendix B for readers less familiar with the area.

### 3.2 Diversity-based Subset Selection

Due to limited computing resources, properly selecting samples instead of using all samples to fit a model is more efficient in problems with a large sample set. In continual learning, this helps overcome the catastrophic forgetting of previously seen data when faced with online data streams. Suppose that we have a model fitted on observed samples $\mathcal{D} \triangleq \{(x_1, y_1), \ldots, (x_t, y_t)\}$, where $x_i \in \mathcal{X}$ and $y_i$ is the corresponding observation. In the context of subset selection, our goal is to maintain a fixed-size retained subset for subset-fitted GP updates as new observations arrive. Let $\mathbf{g}_t$ denote the sample embedding associated with the sample observed at time $t$; in GSSBO this ~~will be the sensitivity embedding defined below~~ is the sensitivity embedding formally defined in Eq. 3 of Section 4. Inspired by the gradient-diversity idea of Aljundi et al. (2019), we adopt a

diversity-oriented subset-selection surrogate and instantiate it with GP-specific sensitivity embeddings rather than the parameter-gradient constraints used in continual learning. To solve the constraint, we use the geometric properties of these embeddings. Directly optimizing the solid angle subtended by their directions is computationally expensive. Following the gradient-diversity surrogate in (Aljundi et al., 2019), we instead maximize the dispersion of the normalized embedding directions in the fixed-size buffer. This encourages diversity in the embedding space, which we use as a surrogate for reducing redundancy among retained observations. How to determine the buffer size will be detailed in Section 4.3. The previous problem thus becomes a surrogate for selecting a subset $\mathcal{U}$ of the samples that maximizes the diversity of their embeddings:

$$\mathrm{Var}_{i \in \mathcal{U}} \left[ \frac{\mathbf{g}_i}{\|\mathbf{g}_i\|} \right] = 1 - \frac{1}{|\mathcal{U}|^2} \sum_{i,j \in \mathcal{U}} \frac{\langle \mathbf{g}_i, \mathbf{g}_j \rangle}{\|\mathbf{g}_i\| \, \|\mathbf{g}_j\|}. \tag{1}$$

In the GSSBO instantiation, $|\mathcal{U}| = M$, where $M$ denotes the buffer size, and $\mathbf{g}_i/\|\mathbf{g}_i\|$ denotes the normalized embedding direction of sample $i$. The larger the value of this formula, the more dispersed the selected embedding directions are; we therefore interpret it as an embedding-space diversity objective for the retained subset. This empirical surrogate objective is agnostic to how the sample embeddings are computed, making it straightforward to integrate into subset-based methods.

## 4 Bayesian Optimization with Gradient-based Sample Selection

### 4.1 Response-Gradient Scores and Sensitivity Embeddings

In the previous section, we introduced a diversity-based subset-selection objective. In the GP setting, we do not assume access to derivatives of the black-box objective with respect to the input $x_i$. Instead, we derive a response-gradient score from the GP log marginal likelihood with respect to the observations: $s_i = \frac{\partial}{\partial y_i} \log p(\mathbf{y} \mid \mathbf{X}, \theta)$. This scalar measures how the fitted log-likelihood responds when the $i$-th response is perturbed, and serves as the starting point for the sensitivity embedding used below for subset maintenance. In a GP model, given a set of samples $\mathcal{D} = \{(x_1, y_1), \ldots, (x_n, y_n)\}$, let $\mathbf{K}$ be the kernel Gram matrix with entries $K_{ij} = k(x_i, x_j; \theta)$, and define the noisy observation covariance $\mathbf{K}_y \triangleq \mathbf{K} + \sigma_n^2 I$, where $\theta$ denotes the kernel hyperparameters and $\sigma_n^2$ is the observation-noise variance. Assuming that the observed responses follow a multivariate normal distribution with mean $\boldsymbol{\mu}$ and covariance $\mathbf{K}_y$, the probability density function is $p(\mathbf{y} \mid \mathbf{X}, \theta) = \frac{1}{(2\pi)^{n/2} |\mathbf{K}_y|^{1/2}} \exp\left(-\frac{1}{2}(\mathbf{y} - \boldsymbol{\mu})^\top \mathbf{K}_y^{-1} (\mathbf{y} - \boldsymbol{\mu})\right)$. Taking the logarithm of it, we derive the log-likelihood function:

$$\log p(\mathbf{y} \mid \mathbf{X}, \theta) = -\frac{1}{2}(\mathbf{y} - \boldsymbol{\mu})^\top \mathbf{K}_y^{-1} (\mathbf{y} - \boldsymbol{\mu}) - \frac{1}{2} \log |\mathbf{K}_y| - \frac{n}{2} \log(2\pi). \tag{2}$$

*Remark* 4.1. The log-likelihood function in equation 2 comprises three terms. The first term, $-\frac{1}{2}(\mathbf{y} - \boldsymbol{\mu})^\top \mathbf{K}_y^{-1}(\mathbf{y} - \boldsymbol{\mu})$, represents the sample fit under the observation covariance $\mathbf{K}_y$. The second term, $-\frac{1}{2} \log |\mathbf{K}_y|$, penalizes model complexity through the log-determinant of the covariance matrix. The third term, $-\frac{n}{2} \log(2\pi)$, is constant with respect to the parameters and thus does not affect the gradient calculation.

The derivative of the log marginal likelihood with respect to $\mathbf{y}$ describes how the fitted GP model responds to perturbations in the observed responses. The derivatives in this subsection are partial derivatives with the kernel hyperparameters, mean function, and observation-noise variance fixed at their current fitted values; the sensitivity embeddings do not differentiate through hyperparameter re-estimation. Throughout, "gradient-based" refers to these response-score derivatives of the GP log marginal likelihood, not to derivatives of the black-box objective $f$. Since the second and third terms in equation 2 do not depend on $\mathbf{y}$, we define the response-gradient vector as $\mathbf{s} \triangleq \frac{\partial \log p(\mathbf{y} \mid \mathbf{X}, \theta)}{\partial \mathbf{y}} = -\mathbf{K}_y^{-1}(\mathbf{y} - \boldsymbol{\mu})$. We denote its $i$-th component by $s_i = -\left(\mathbf{K}_y^{-1}(\mathbf{y} - \boldsymbol{\mu})\right)_i$. Proposition 5.3 later relates the discarded entries of this scalar score vector to the posterior-mean distortion caused by fitting on a retained subset. Because $\mathbf{K}_y^{-1}$ is generally dense, this scalar quantity is evaluated in the context of the full observation covariance of the observed dataset.

~~Because the scalar score $s_i$ alone is not a vector representation for diversity-based subset-selection,~~ Because the scalar score $s_i$ does not provide a non-degenerate multidimensional geometry for cosine-diversity selection—after normalization, the cosine similarity of scalars reduces to a sign comparison—we use the sensitivity of

the full response-gradient vector to the $i$-th response:

$$\mathbf{g}_i \triangleq \frac{\partial \mathbf{s}}{\partial y_i} = -\mathbf{K}_y^{-1}\mathbf{e}_i, \tag{3}$$

where $\mathbf{e}_i$ is the $i$-th standard basis vector. We call $\mathbf{g}_i$ the sensitivity embedding of sample $i$: it is the sensitivity of the full response-gradient vector to perturbing $y_i$. Equivalently, $\mathbf{g}_i$ has the precision-column form $-\mathbf{K}_y^{-1}\mathbf{e}_i$, the negative $i$-th column of the observation precision matrix $\mathbf{K}_y^{-1}$; whereas the scalar score $s_i$ depends on the residual $\mathbf{y} - \boldsymbol{\mu}$ through $\mathbf{K}_y^{-1}(\mathbf{y} - \boldsymbol{\mu})$, the embedding $\mathbf{g}_i$ is determined by the corresponding precision column. We use these embeddings to promote diversity in the retained subset.

**Relation between the scalar scores and the sensitivity embeddings.** The two objects are distinct but exactly related. Writing $\mathbf{z} = \mathbf{y} - \boldsymbol{\mu}$ for the centered responses, the score vector and the embeddings are algebraically linked through $\mathbf{s} = -\mathbf{K}_y^{-1}\mathbf{z} = \sum_i z_i\,\mathbf{g}_i$, and $s_i = \mathbf{g}_i^\top\mathbf{z}$, so $\mathbf{s}$ is the response-weighted combination of the embeddings, and each scalar score is the inner product of the centered responses with the corresponding embedding.

The two representations carry different information. $\mathbf{g}_i = \partial\mathbf{s}/\partial y_i$ is the $i$-th column of the response-score Jacobian—the Hessian of the log marginal likelihood in $\mathbf{y}$, constant because the log-likelihood is quadratic in $\mathbf{y}$—which is why the embeddings carry no first-order response information. The embeddings $\{\mathbf{g}_i\}$ are the (negative) columns of the observation precision matrix and are therefore *response-independent*: conditional on the selection pool and the current fitted hyperparameters, they contain no explicit dependence on the centered responses: they encode the conditional-dependence structure of the noisy responses (indexed by the observed inputs), and can be viewed as a pool-specific precision-column feature representation of the samples. GSSBO promotes directional dispersion in this precision-embedding space; the selection step is thus a diversity rule in the embedding geometry rather than a rule driven by the realized responses. The scalar scores, in contrast, incorporate the realized residuals but are one-dimensional: cosine similarity between them reduces to a sign comparison, which induces a *sign-balanced* selection rule that we evaluate as a variant (Appendix C.3). They admit an exact model-based interpretation: $s_i = -(y_i - \mu_{-i}(x_i))/v_{-i}^{(y)}(x_i)$, where $\mu_{-i}$ is the leave-one-out predictive mean of observation $i$ under the full-data fit and $v_{-i}^{(y)}(x_i) = \mathrm{Var}\big(y_i \mid \mathcal{D} \setminus \{(x_i, y_i)\}\big)$ is the leave-one-out predictive variance of the noisy observation, i.e., including the observation-noise term (Williams and Rasmussen, 2006, Sec. 5.4.2); thus $|s_i|$ measures how poorly the remaining data explain observation $i$, and $\mathrm{sign}(s_i)$ records the direction of that error.

The scalar scores are also the quantities that enter the theory: Proposition 5.3 shows that the discarded scalar scores $s_\mathcal{R}$, together with the conditional cross-covariance residual $c_\mathcal{U}(x)$, determine the posterior-mean distortion of a fixed retained subset. The embeddings are what the algorithm uses to choose the subset; the scalar scores are what the theory uses to assess a chosen subset. The algebraic relation links the two objects but does not by itself imply that either selection rule controls the posterior gap.

More generally, when the embedding is computed on a finite ambient set $\mathcal{A}$, we write

$$\mathbf{g}_i^{(\mathcal{A})} = -\big(\mathbf{K}_y^{(\mathcal{A})}\big)^{-1}\mathbf{e}_i^{(\mathcal{A})}, \qquad i \in \mathcal{A}, \tag{4}$$

where $\mathbf{K}_y^{(\mathcal{A})} = \mathbf{K}_{\mathcal{A}\mathcal{A}} + \sigma_n^2 I_{|\mathcal{A}|}$ is the observation covariance restricted to $\mathcal{A}$, and $\mathbf{e}_i^{(\mathcal{A})} \in \mathbb{R}^{|\mathcal{A}|}$. When the ambient set is clear from context, we write $\mathbf{g}_i$. After obtaining the sensitivity embedding for each sample, we can use the directional dispersion of these normalized embedding directions as an alternative diversity objective, i.e. promoting embedding-direction diversity in the retained subset. Constructing these embeddings requires access to the candidate observations' inverse observation-covariance structure and introduces additional overhead beyond the subset-GP refit. We account for this overhead separately in the complexity summary below.

---

The Gram matrix of the embeddings is $\mathbf{G}^\top\mathbf{G} = \mathbf{K}_y^{-2}$, so the cosine rule measures angles in the squared-precision geometry rather than in the original kernel $k$ or the precision $\mathbf{K}_y^{-1}$ itself.

## 4.2   Gradient-based Sample Selection

As the number of observed samples increases, fitting a GP model can become prohibitively expensive, especially in large-scale scenarios. A common remedy is to work with a subset of samples of size $M \ll N$, thereby reducing the computational cost of GP updates. The efficiency and effectiveness of GP model fitting in BO are closely related to the quality of the chosen subset. This raises the question: *How do we choose a compact retained subset that is useful for subset-fitted GP updates?* Inspired by the success of gradient-based subset selection methods in machine learning, we use GP response-sensitivity embeddings to guide the maintenance of such subsets within BO. To this end, we introduce a sensitivity-embedding-based subset-maintenance methodology that aims to improve retained-subset coverage within a limited sample buffer size. By using sensitivity embeddings, the approach seeks to reduce computational burden while empirically maintaining surrogate quality as the sample set grows. We begin by modeling the objective function $f$ with a GP and setting a buffer size $M$. Initially, the algorithm observes $f$ at $n_0$ samples and uses all initial observations before subset maintenance is activated. After each subsequent evaluation, if the number of samples exceeds $M$, we perform a sensitivity-embedding-based subset selection step to keep a subset of size $M$ for the next GP update.

## 4.3   Gradient-based Sample Selection BO

The following outlines the GSSBO implementation details and practical considerations. *We highlight the key insight of this subsection: we tackle the scalability of BO by maintaining a compact subset selected by the proposed subset-maintenance heuristic.*

**Algorithmic Overview.** Algorithm 1 summarizes the online subset-maintenance procedure. In each iteration, GSSBO selects a new point using the current GP posterior and acquisition function, evaluates the corresponding observation, and updates the dataset with the new sample. If $|\mathcal{D}| \leq M$, the GP is updated using all observed samples. Once the buffer limit is exceeded, the method switches to subset maintenance: the initial-design samples and the newly acquired sample $(x_t, y_t)$ are retained (the *retained-initial-design* default), and the remaining $M - n_0 - 1$ samples are selected from a selection pool $\mathcal{P}_t \subseteq \mathcal{D}_t$ according to the cosine-diversity rule over normalized sensitivity embeddings in Eq. 1, and the GP surrogate is refit on the resulting subset $\mathcal{U}_t$. For a finite ambient set $\mathcal{A}$, the sensitivity embedding in Eq. 3 is computed using the observation covariance restricted to $\mathcal{A}$; taking $\mathcal{A} = \mathcal{P}_t$ gives the pool-specific embeddings used by the selection step, while $\mathcal{A} = \mathcal{D}_t$ recovers the full-history embedding definition.

**Greedy cosine-diversity selection.**   Concretely, the selection step in Algorithm 1 is implemented greedily. Given the pool-specific embeddings, we normalize $\mathbf{v}_i = \mathbf{g}_i^{(\mathcal{P}_t)}/\|\mathbf{g}_i^{(\mathcal{P}_t)}\|_2$ and form pairwise scores $C_{ij} = \mathbf{v}_i^\top \mathbf{v}_j$. The initial design and the latest observation are inserted first; the remaining $M - n_0 - 1$ samples are chosen by repeatedly adding the candidate with the smallest cumulative cosine similarity $h_i = \sum_{j \in \mathcal{S}} C_{ij}$ to the current retained set $\mathcal{S}$, with $h_i$ updated incrementally after each insertion. Given the pool-level cosine scores, the greedy stage costs $\mathcal{O}(q_t M)$; embedding construction and cosine-score construction are accounted for separately in the overhead terms $C_{\text{emb}}$ and $C_{\text{div}}$ below. Appendix C.1 gives the full details.

In our implementation, the selection pool is the full history, $\mathcal{P}_t = \mathcal{D}_t$; the pool-specific embeddings are recomputed at each selection step via one Cholesky factorization of $\mathbf{K}_y^{(\mathcal{P}_t)}$, and this cost is included in all reported runtimes. Appendix C.6 varies both choices—restricted pools with window size $|\mathcal{W}_t| \in \{M, 2M, 4M\}$ and a cached rank-one embedding refresh—and reports the effect on regret and runtime.

**(1) Dynamic Buffer Size.** In practice, a suitable buffer size may be difficult to choose in advance. We therefore use a simple runtime-triggered rule to initialize $M$. Let $\bar{T}$ be the average wall-clock time of the initial iterations and $T_{\text{current}}$ the current iteration time. When $T_{\text{current}}$ first exceeds the user-specified threshold $Z \times \bar{T}$, we set $M = |\mathcal{D}|$ at that iteration and keep this value fixed thereafter. In this way, the transition itself does not discard previously collected observations, while subsequent iterations proceed with subset maintenance under the fixed buffer size. This rule provides a simple way to balance computational cost and retained-data coverage without requiring $M$ to be chosen a priori.

**(2) Retaining Latest Observations.** During the procedure, the newly acquired sample, $(x_t, y_t)$, is always included in the subset. This makes the most recent observation available to the next subset-fitted GP update, which can help the retained subset track newly explored regions. Additionally, this design addresses a practical limitation of fixed-size representations in BO (McIntire et al., 2016): a constrained representation size may otherwise delay the use of new observations in the model. It may also reduce repeated refitting on subsets concentrated around the same current best region, since the most recent observation is always retained in the next iteration.

Here we highlight the difference between our method and SparseGP. SparseGP methods introduce inducing-point approximations together with additional posterior-correction mechanisms. In contrast, our method maintains a compact subset of observed samples via the subset-maintenance heuristic and refits the GP only on that retained subset. The computational gain in GSSBO therefore comes from subset-of-data GP updates rather than from inducing-point posterior corrections.

---

**Algorithm 1** Gradient-based Sample Selection BO

---

1: **Initialization:** Obtain $n_0$ initial samples $\mathcal{D} = \{(x_i, y_i)\}_{i=1}^{n_0}$, and fit an initial GP model. Set buffer size $M > n_0$, total budget $N$, average initial iteration time $\bar{T}$, and threshold factor $Z$. Initialize `switched` ← `false`.
2: **for** $t = n_0 + 1$ to $N$ **do**
3:     Select $x_t = \arg\max_x \alpha(x; p_t(f))$, where $p_t(f)$ is the current fitted GP posterior and $\alpha$ is the acquisition function.
4:     Evaluate $y_t = f(x_t)$ and set $\mathcal{D} \leftarrow \mathcal{D} \cup \{(x_t, y_t)\}$.
5:     **if** not switched **then**
6:         Let $T_{\text{current}}$ be the current iteration time.
7:         **if** $T_{\text{current}} > Z \times \bar{T}$ **then**
8:             Set `switched` ← `true`,   $M \leftarrow |\mathcal{D}|$.
9:         **end if**
10:     **end if**
11:     **if** switched **then**
12:         Define the forced set $\mathcal{F}_t = \{(x_i, y_i)\}_{i=1}^{n_0} \cup \{(x_t, y_t)\}$. Choose a discretionary candidate set $\mathcal{W}_t \subseteq \mathcal{D}_t$ (e.g. a recent window) and set the selection pool $\mathcal{P}_t = \mathcal{F}_t \cup \mathcal{W}_t$, with $|\mathcal{P}_t| \geq M$ and $q_t = |\mathcal{P}_t|$.
13:         Compute or update sensitivity embeddings $\mathbf{g}_i^{(\mathcal{P}_t)}$ for all samples in $\mathcal{P}_t$.
14:         Initialize the subset $\mathcal{U}_t \leftarrow \mathcal{F}_t$.
15:         Select the remaining $M - |\mathcal{F}_t|$ samples from $\mathcal{P}_t \setminus \mathcal{F}_t$ according to the cosine-diversity rule in Eq. 1, using these pool-specific embeddings, and add them to $\mathcal{U}_t$.
16:         Update the subset-fitted GP using the $M$ samples in $\mathcal{U}_t$.
17:     **else**
18:         Update the GP using all samples in $\mathcal{D}$.
19:     **end if**
20: **end for**

---

The dominant computational trade-off in GSSBO is between refitting the GP on a retained subset of size $M$ and maintaining that subset over time. At iteration $t$, let $n_t = |\mathcal{D}_t|$ be the number of accumulated observations and let $q_t = |\mathcal{P}_t|$ be the size of the selection pool. In standard GP-UCB, the full-data GP refit/factorization cost is $\mathcal{O}(n_t^3)$. Once subset maintenance is activated, the subset-fitted GP update in GSSBO uses $M$ retained observations, so the GP refit/factorization term is $\mathcal{O}(M^3)$ for fixed hyper-parameters. This term should be read as the subset-GP refitting cost after the switch, not as a complete end-to-end per-iteration bound for GSSBO. The total update also includes the cost of refreshing the sensitivity embeddings, $C_{\text{emb}}(n_t, q_t)$, and the cost of the cosine-diversity subset-maintenance step, $C_{\text{div}}(q_t, M)$. Thus, compared with the $\mathcal{O}(n_t^3)$ refit/factorization term of full GP-UCB, GSSBO caps the subset-GP refit term at $\mathcal{O}(M^3)$ after the switch while making the additional subset-maintenance overhead explicit. RSSBO has the same subset-GP refit term but uses random subset maintenance instead of the sensitivity-embedding and cosine-diversity step.

The terms $C_{\text{emb}}$ and $C_{\text{div}}$ depend on the embedding-maintenance and cosine-diversity implementation. If exact full-data precision-column embeddings are recomputed over all accumulated observations, then $q_t = n_t$

and the precision-structure construction is dominated by $\mathcal{O}(n_t^3)$ factorization. If the observation precision matrix is maintained under fixed hyper-parameters using a rank-one block update, the refresh can be $\mathcal{O}(n_t^2)$ per new observation with $\mathcal{O}(n_t^2)$ memory; when hyper-parameters are refit, this cache may need to be rebuilt. Given cached pairwise cosine scores, a greedy cosine-diversity implementation costs $\mathcal{O}(q_t M)$; without cached similarities, forming all pairwise cosine scores costs $\mathcal{O}(d_{g,t} q_t^2)$, where $d_{g,t}$ is the embedding dimension for that iteration. For pool-specific precision-column embeddings, $d_{g,t} = q_t$, so naive pairwise cosine construction costs $\mathcal{O}(q_t^3)$; for full-history embeddings evaluated on $q_t$ pool candidates, $d_{g,t} = n_t$. Restricted-pool implementations correspond to smaller $q_t$, while the full-history case is obtained by taking $\mathcal{W}_t = \mathcal{D}_t$, i.e. $\mathcal{P}_t = \mathcal{D}_t$. Since a restricted pool bounds only the discretionary candidates $\mathcal{W}_t$, $q_t = |\mathcal{F}_t \cup \mathcal{W}_t|$ counts any forced points lying outside the candidate window.

## 5 Theoretical Analysis

We analyze the posterior distortion caused by fitting the GP on a retained subset $\mathcal{U}$ rather than on the full dataset $\mathcal{D}$. Throughout this section and the appendix, vectors and matrices are not boldfaced; vector norms are Euclidean and matrix norms are operator norms. We use the standard noisy GP model with observation-noise variance $\sigma_n^2 > 0$, write the results in zero-mean form, and compare full-data and subset-fitted posteriors under shared kernel hyper-parameters and observation-noise variance. The results are conditional diagnostics for a fixed retained subset, not unconditional guarantees for the cosine-diversity selection rule.

Fix a partition $\mathcal{D} = \mathcal{U} \cup \mathcal{R}$, where $\mathcal{U}$ is retained and $\mathcal{R}$ is discarded. Define $A = K_{\mathcal{U}\mathcal{U}} + \sigma_n^2 I$, $B = K_{\mathcal{U}\mathcal{R}}$, $C = K_{\mathcal{R}\mathcal{R}} + \sigma_n^2 I$, and $S = C - B^\top A^{-1} B$. For a test point $x$, let $k_{\mathcal{U}}(x) = K_{\mathcal{U}x}$, $k_{\mathcal{R}}(x) = K_{\mathcal{R}x}$, $r_{\mathcal{U}} = y_{\mathcal{R}} - B^\top A^{-1} y_{\mathcal{U}}$, and $c_{\mathcal{U}}(x) = k_{\mathcal{R}}(x) - B^\top A^{-1} k_{\mathcal{U}}(x)$. Let $(\mu_{\mathcal{D}}, \sigma_{\mathcal{D}}^2)$ and $(\mu_{\mathcal{U}}, \sigma_{\mathcal{U}}^2)$ be the full-data and retained-subset posteriors. Here $m = |\mathcal{U}|$ and $r = |\mathcal{R}|$, so $A \in \mathbb{R}^{m \times m}$, $B \in \mathbb{R}^{m \times r}$, $C, S \in \mathbb{R}^{r \times r}$, $k_{\mathcal{U}}(x) \in \mathbb{R}^m$, and $k_{\mathcal{R}}(x), c_{\mathcal{U}}(x) \in \mathbb{R}^r$.

**Posterior gap.** The following theorem gives exact posterior-gap identities for this fixed partition.

**Theorem 5.1** (Posterior gap for a fixed subset-fitted GP under shared hyper-parameters). *With the notation above, for every $x$, $\mu_{\mathcal{D}}(x) - \mu_{\mathcal{U}}(x) = c_{\mathcal{U}}(x)^\top S^{-1} r_{\mathcal{U}}$ and $\sigma_{\mathcal{U}}^2(x) - \sigma_{\mathcal{D}}^2(x) = c_{\mathcal{U}}(x)^\top S^{-1} c_{\mathcal{U}}(x) \geq 0$. Consequently, $|\mu_{\mathcal{D}}(x) - \mu_{\mathcal{U}}(x)| \leq \|c_{\mathcal{U}}(x)\| \, \|S^{-1}\| \, \|r_{\mathcal{U}}\|$ and $0 \leq \sigma_{\mathcal{U}}^2(x) - \sigma_{\mathcal{D}}^2(x) \leq \|c_{\mathcal{U}}(x)\|^2 \, \|S^{-1}\|$. Since $S \succeq \sigma_n^2 I$, the coarser bounds $|\mu_{\mathcal{D}}(x) - \mu_{\mathcal{U}}(x)| \leq \|c_{\mathcal{U}}(x)\| \, \|r_{\mathcal{U}}\| / \sigma_n^2$ and $\sigma_{\mathcal{U}}^2(x) - \sigma_{\mathcal{D}}^2(x) \leq \|c_{\mathcal{U}}(x)\|^2 / \sigma_n^2$ also hold.*

*Remark* 5.2 (Interpretation). Theorem 5.1 shows that posterior distortion is governed by the discarded-response residual $r_{\mathcal{U}}$ and the conditional cross-covariance residual $c_{\mathcal{U}}(x)$. Small residuals imply a small subset/full posterior gap, but the theorem does not prove that the GSSBO cosine-diversity rule always makes these residuals small.

The next proposition connects the scalar response-gradient score used in GSSBO's diagnostics with the posterior-mean distortion above.

**Proposition 5.3** (Residual bridge for scalar response scores). *Under the assumptions and notation of Theorem 5.1, let $z_{\mathcal{D}}$ denote the response vector centered by the chosen GP mean function. Under the zero-mean notation of the theorem, $z_{\mathcal{D}} = y_{\mathcal{D}}$. Define $s = -(K_{\mathcal{D}\mathcal{D}} + \sigma_n^2 I)^{-1} z_{\mathcal{D}}$, and let $s_{\mathcal{R}}$ be the entries indexed by $\mathcal{R}$. Then $s_{\mathcal{R}} = -S^{-1} r_{\mathcal{U}}$ and $r_{\mathcal{U}} = -S s_{\mathcal{R}}$. Consequently, for every $x$, $\mu_{\mathcal{D}}(x) - \mu_{\mathcal{U}}(x) = -c_{\mathcal{U}}(x)^\top s_{\mathcal{R}}$, and $|\mu_{\mathcal{D}}(x) - \mu_{\mathcal{U}}(x)| \leq \|c_{\mathcal{U}}(x)\| \, \|s_{\mathcal{R}}\|$. If $\rho_{\mathcal{U}} := \sup_{x \in \mathcal{X}} \|c_{\mathcal{U}}(x)\| < \infty$, then $\sup_{x \in \mathcal{X}} |\mu_{\mathcal{D}}(x) - \mu_{\mathcal{U}}(x)| \leq \rho_{\mathcal{U}} \|s_{\mathcal{R}}\|$.*

This proposition gives the scalar response scores a posterior-distortion interpretation: the discarded full-data scores determine the posterior-mean distortion of a fixed retained subset through the conditional cross-covariance residual $c_{\mathcal{U}}(x)$.

For acquisition stability, define the full-data and subset-fitted UCB values with the same $\beta_t \geq 0$: $\alpha_{\mathcal{D}}^{\mathrm{UCB}}(x) = \mu_{\mathcal{D}}(x) + \sqrt{\beta_t} \, \sigma_{\mathcal{D}}(x)$ and $\alpha_{\mathcal{U}}^{\mathrm{UCB}}(x) = \mu_{\mathcal{U}}(x) + \sqrt{\beta_t} \, \sigma_{\mathcal{U}}(x)$, where $\sigma_{\mathcal{D}}(x) = \sqrt{\sigma_{\mathcal{D}}^2(x)}$ and $\sigma_{\mathcal{U}}(x) = \sqrt{\sigma_{\mathcal{U}}^2(x)}$.

**Corollary 5.4** (UCB acquisition stability for a fixed retained subset). *Under the assumptions of Theorem 5.1, for every $x \in \mathcal{X}$, $|\alpha_{\mathcal{D}}^{\mathrm{UCB}}(x) - \alpha_{\mathcal{U}}^{\mathrm{UCB}}(x)| \leq \|c_{\mathcal{U}}(x)\| \, \|s_{\mathcal{R}}\| + \sqrt{\beta_t} \, \|c_{\mathcal{U}}(x)\| \sqrt{\|S^{-1}\|}$. If $\rho_{\mathcal{U}} := \sup_{x \in \mathcal{X}} \|c_{\mathcal{U}}(x)\| < \infty$, then $\sup_{x \in \mathcal{X}} |\alpha_{\mathcal{D}}^{\mathrm{UCB}}(x) - \alpha_{\mathcal{U}}^{\mathrm{UCB}}(x)| \leq \rho_{\mathcal{U}} \|s_{\mathcal{R}}\| + \sqrt{\beta_t} \rho_{\mathcal{U}} \sqrt{\|S^{-1}\|}$.*

**Remark.** Theorem 5.1, Proposition 5.3, and Corollary 5.4 are conditional diagnostic and stability statements for a fixed retained subset under shared hyper-parameters and observation-noise variance. They do not

Figure 2: Cumulative regret of algorithms on synthetic benchmarks and the diabetes hyperparameter optimization task. Solid lines denote the mean over 50 runs; shaded regions indicate $\pm 1$ standard deviation.

Table 1: Final cumulative regret at iteration 1000 for the experiments of Figure 2 (mean $\pm$ 95% confidence interval over the 50 runs; lower is better; bold marks the lowest mean in each column).

| Method | Eggholder2 $\times 10^3$ | Hart6 $\times 10^3$ | Levy20 $\times 10^5$ | Powell50 $\times 10^7$ | Rastrigin100 $\times 10^6$ | Diabetes $\times 10^2$ |
|---|---|---|---|---|---|---|
| GP-UCB | $1.196 \pm 0.109$ | $\mathbf{3.540 \pm 0.105}$ | $1.568 \pm 0.065$ | $5.717 \pm 0.292$ | $1.285 \pm 0.008$ | $\mathbf{0.824 \pm 0.045}$ |
| GSSBO | $1.425 \pm 0.121$ | $3.874 \pm 0.119$ | $1.504 \pm 0.071$ | $\mathbf{5.479 \pm 0.280}$ | $\mathbf{1.232 \pm 0.009}$ | $0.891 \pm 0.062$ |
| RSSBO | $2.020 \pm 0.184$ | $5.677 \pm 0.153$ | $1.583 \pm 0.071$ | $5.751 \pm 0.296$ | $1.292 \pm 0.008$ | $1.322 \pm 0.085$ |
| VecchiaBO | $\mathbf{0.906 \pm 0.086}$ | $3.654 \pm 0.345$ | $1.599 \pm 0.076$ | $6.154 \pm 0.315$ | $1.381 \pm 0.009$ | $0.914 \pm 0.036$ |
| LR-First $m$ | $1.916 \pm 0.107$ | $7.441 \pm 0.196$ | $1.782 \pm 0.082$ | $6.274 \pm 0.321$ | $1.454 \pm 0.010$ | $1.125 \pm 0.074$ |
| SparseGP | $1.669 \pm 0.155$ | $4.770 \pm 0.116$ | $1.850 \pm 0.085$ | $7.347 \pm 0.376$ | $1.732 \pm 0.011$ | $1.581 \pm 0.107$ |
| IPA | $1.529 \pm 0.136$ | $4.584 \pm 0.127$ | $1.589 \pm 0.073$ | $6.362 \pm 0.325$ | $1.680 \pm 0.011$ | $1.158 \pm 0.083$ |
| mini-META | $1.460 \pm 0.123$ | $4.193 \pm 0.116$ | $\mathbf{1.498 \pm 0.069}$ | $6.036 \pm 0.309$ | $1.664 \pm 0.011$ | $1.236 \pm 0.084$ |

prove that the GSSBO selection rule necessarily makes $\|s_{\mathcal{R}}\|$, $\rho_{\mathcal{U}}$, $\|r_{\mathcal{U}}\|$, or $\|S^{-1}\|$ small, nor do they provide cumulative-regret, argmax-equivalence, or unconditional optimality guarantees. The experiments evaluate downstream optimization performance, runtime, and an auxiliary RMSE diagnostic for the retained-subset GP fit. A direct scalar-versus-vector ablation (Appendix C.3) evaluates the practical representation choice rather than validating the bound; the remaining theory–practice gap is that neither the vector cosine-diversity rule nor the scalar-score variant is shown to control the residual quantities or acquisition-level error. Appendix C.4 reports direct measurements of these residual and posterior-gap quantities for the default vector rule against matched random selection, as empirical diagnostics under the stated protocol rather than guarantees. Full proofs are deferred to the appendix.

**Remark (hyperparameter re-estimation).** The analysis above compares the full-data and subset-fitted posteriors under shared kernel hyperparameters and observation-noise variance; this deliberately isolates the effect of data pruning itself (Appendix A.7). In the experiments of Section 6, hyperparameters are re-estimated by maximum likelihood at every iteration on the current retained subset, so the reported behavior combines the pruning effect analyzed here with a hyperparameter re-estimation effect that the theory does not cover. The bounds should therefore be read as diagnostics evaluated at fixed hyperparameters. Appendix C.7 quantifies the re-estimation effect empirically: freezing hyperparameters at the buffer switch moderately increases final cumulative regret on both tested tasks relative to per-iteration re-estimation.

## 6   Experiments

In this section, we conduct numerical experiments to evaluate the practical efficiency of GSSBO. The objective of the numerical experiments is threefold: (1) to evaluate computational efficiency; (2) to assess optimization performance; and (3) to report an auxiliary RMSE diagnostic for the retained-subset GP fit. To assess the performance of our proposed methods, we test five benchmark functions (Eggholder2, Hart6, Levy20, Powell50, Rastrigin100) and a real-world diabetes hyperparameter optimization task.

**Experimental Setup.** We choose UCB as the acquisition function in GSSBO, and compare GSSBO with the following benchmarks: *(1) Standard GP-UCB* (Srinivas et al., 2009); *(2) Random Sample Selection GP-UCB (RSSBO, our ablation)*, which mirrors our approach in restricting the sample set size but chooses

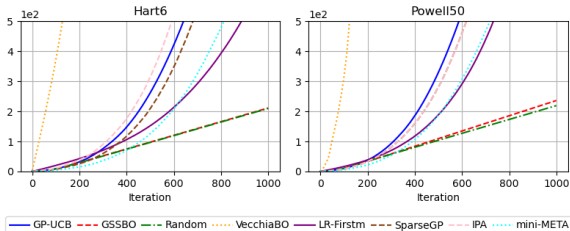

Figure 3: Cumulative time cost of algorithms.

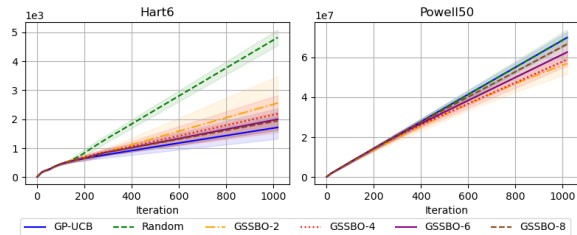

Figure 4: Sensitivity analysis of $Z$.

which samples randomly (it retains the latest observation, as GSSBO does, but unlike GSSBO's default it does not retain the initial design); *(3) VecchiaBO* (Jimenez and Katzfuss, 2023), which utilizes the Vecchia approximation method in BO; *(4) LR-First $m$* (Williams and Seeger, 2000), which uses a low-rank approximation based on the first $m$ samples. *(5) SparseGP* (Lawrence et al., 2002), which introduces a small number of inducing points to obtain a sparse approximation of GP; *(6) IPA* (Moss et al., 2023), which multiplies an expected-improvement-based quality function by the GP posterior and selects inducing points regarding quality-diversity; and *(7) mini-META* (Calandriello et al., 2022), which scales GP optimization by repeatedly evaluating each selected point until its posterior uncertainty falls below a preset threshold. (Additional baselines and high-dimensional tasks are reported in the appendix.) We employ a Matérn 5/2 kernel for the GP, with hyperparameters learned via maximum likelihood estimation. Both GSSBO and RSSBO use the same buffer size $M$, dynamically adjusted by a parameter $Z = 4$. $M$ is usually around 100 and will change with $Z$. The size of the initial set is 20. Each experiment is repeated 50 times, and the total number of iterations is 1000. All experiments were conducted on a MacBook Pro with Apple M2 Pro (10-core CPU, 16 GB unified RAM). For maximization tasks, we report cumulative regret over a horizon $T$ (with $T = N$ in the experiments below, where $N$ is the total budget in Algorithm 1): $R_T = \sum_{t=1}^{T}\big(f(x^\star) - f(x_t)\big)$. For the real-world hyperparameter optimization task, where the raw objective is a validation classification error to be minimized, we equivalently maximize its negative so that the same regret convention applies.

## 6.1 Synthetic Test Problems

**Computational Efficiency Analysis.** Figure 3 compares the cumulative runtime (in seconds) over 1000 iterations on low-dimensional Hart6 and high-dimensional Powell50 (results for other test functions are provided in the appendix due to space constraints). In both plots, VecchiaBO incurs a rapidly accelerating runtime, whereas GSSBO and RSSBO remain notably lower than GP-UCB. While VecchiaBO reduces the cost of GP fitting by conditioning on nearest neighbors, its runtime is dominated by the costly maintenance of a structured neighbor graph, which scales poorly with dimensionality and sample size. The running speeds of LR-First $m$, SparseGP, and mini-META are all improved compared to the standard GP-UCB. IPA has no obvious advantage in terms of time consumption because it needs to calculate the sample quality. In contrast, GSSBO and RSSBO have much lower runtime in these experiments. Once the active subset is restricted to size $M \ll n$, the subset-GP refit component in GSSBO is capped at $\mathcal{O}(M^3)$, while the total runtime also includes the implemented sensitivity-embedding construction and cosine-diversity subset-selection overhead described above. GSSBO and RSSBO are often similar in runtime, though GSSBO can be slightly higher due to the overhead of the subset-maintenance heuristic. By iteration 1000, the cumulative runtime of GSSBO on both Hart6 and Powell50 is about 10% of that of standard GP-UCB. This empirical comparison suggests that GP refitting is a major computational burden in the large-budget setting considered here. Under this setup, the runtime gap may become larger as $n$ increases.

**Optimization Performance Analysis.** Figure 2 compares methods on multiple functions, evaluating cumulative regret. Overall, GSSBO achieves cumulative regret comparable to standard GP-UCB and lower than RSSBO in most reported settings. In low-dimensional problems such as Eggholder2 and Hart6, GSSBO has a small gap with standard GP-UCB and VecchiaBO while remaining competitive with the other baselines in Figure 2. In high-dimensional settings, such as on Levy20, Powell50, and Rastrigin100, GSSBO obtains among the lowest mean cumulative regrets among the compared methods in Figure 2. Table 1 reports the

corresponding final values (mean $\pm$ 95% confidence interval over the 50 runs). Approximate two-sample tests computed from these summary statistics ($n$=50 per method) show that GSSBO's final cumulative regret is significantly lower than RSSBO's on Eggholder2 ($p \approx 10^{-7}$), Hart6, Rastrigin100, and the diabetes task (all $p < 10^{-15}$), with no significant difference on Levy20 and Powell50. Relative to full-data GP-UCB, GSSBO is significantly lower on Rastrigin100, not significantly different on Levy20, Powell50, and the diabetes task, and significantly higher on Eggholder2 ($p = 0.006$) and Hart6 ($p = 4 \times 10^{-5}$). These outcomes are unchanged under a Holm correction across the twelve comparisons. From the experimental results, the cumulative regret of GSSBO remains competitive over the tested horizon. In particular, GSSBO achieves these results with a substantial reduction in computation time in our experiments, as shown in Figure 3. These results show a favorable empirical trade-off between regret and runtime in the reported settings. A plausible explanation for the stronger performance of GP-UCB in Figure 2 is that it uses full-data GP updates, whereas the other baselines rely on approximation or subset-restriction mechanisms that may reduce surrogate fidelity in some tasks.

**Sensitivity analysis of Hyperparameter $Z$.** We further examine how the dynamic buffer parameter $Z$ affects GSSBO. Figure 4 presents results on two functions: Hart6 and Powell50. For RSSBO, $Z$ remains fixed at 4, whereas for GSSBO, we vary $Z \in \{2, 4, 6, 8\}$. On Hart6, larger $Z$ tends to move GSSBO closer to standard GP-UCB, while RSSBO has higher cumulative regret in these runs. In contrast, in Powell50, smaller $Z$ gives slightly better GSSBO performance in this experiment, which is consistent with a possible benefit of more aggressive subset updates in this setting. In these two cases, Hart6 benefits from a larger $Z$, whereas Powell50 performs slightly better with more aggressive subset limiting. This suggests that the preferred $Z$ may depend on the problem structure. We use $Z = 4$ in the main experiments as a common reference setting, not as a universally optimal choice. The parameter determines when the buffer size $M$ is initialized: a larger $Z$ delays the switch, typically yielding a larger buffer closer to full-GP behavior at higher cost. Given this task-dependent behavior, $Z$ should be treated as a computational trade-off parameter; the fixed-$M$ experiments below provide a hardware-independent alternative.

**Deterministic fixed-buffer results.** Because the runtime-triggered rule depends on wall-clock time, we also report deterministic fixed-buffer experiments. At a fixed buffer size $M$=100 with matched retention, the vector rule attains $3938 \pm 134$ final cumulative regret versus $4972 \pm 92$ for random selection on Hart6, and $1676 \pm 95$ versus $1977 \pm 68$ on Eggholder2; the full selection-rule $\times$ retention grid and the fixed buffers $M \in \{50, 100, 200\}$ are reported in Tables 5 and 4.

## 6.2 Real-World Application

To assess the applicability of GSSBO beyond analytic test functions, we also consider hyperparameter optimization for a diabetes-detection model built from the UCI diabetes dataset (Dua and Graff, 2017). In this task, each BO query $(x_t, y_t)$ corresponds to training the model with a proposed hyperparameter configuration and observing its validation classification error. The search space contains four hyperparameters: batch size in $[32, 128]$, learning rate in $[10^{-6}, 1.0]$, ~~learning-rate decay~~ the optimizer weight-decay coefficient in $[10^{-6}, 1.0]$, and hidden dimension in $[1, 8]$. Concretely, the task uses the Pima Indians Diabetes data (768 samples, 8 features), split 75/25 into training and validation sets with a fixed seed; features are min–max normalized with the scaler fit on the training split. Each query trains a single-hidden-layer MLP (ReLU activation) with the proposed hidden dimension using RMSprop with the proposed learning rate, weight-decay coefficient and batch size for five epochs, and returns the validation classification error. Because network initialization and mini-batch shuffling are not fixed across queries, the objective is stochastic: repeated evaluations of the same configuration return different values. Since the raw objective is an error to be minimized, we optimize its negative when applying the common maximization-based regret convention described above. This experiment tests whether the subset-maintenance strategy remains useful when the objective evaluations are generated by a learning pipeline rather than by a closed-form synthetic benchmark. The results in Figure 2 show that GSSBO performs favorably relative to the compared baselines on this task while using the same subset-maintenance protocol as in the synthetic experiments. This single four-hyperparameter task is illustrative and does not establish generalization to larger-scale hyperparameter-optimization problems.

## 7 Conclusion

BO is known to be effective for optimization in settings where the objective function is expensive to evaluate. In large-budget scenarios, the use of a full GP model can slow the convergence of BO, leading to poor scaling in these cases. In this paper, we investigated the use of sensitivity-embedding-based subset maintenance to accelerate BO. The results show empirically that the GSSBO-selected retained subset can often support BO performance close to full-data GP-UCB while substantially reducing GP fitting cost in the reported settings. Across the reported synthetic and real-world benchmarks, GSSBO reduces runtime while retaining competitive optimization performance. The sensitivity analysis suggests that the preferred $Z$ can depend on the problem setting. Overall, these findings support response-sensitivity-based subset maintenance as a practical route for addressing BO scaling challenges. Our theoretical results should be read as conditional posterior- and acquisition-stability diagnostics for fixed retained subsets; the empirical results evaluate how the proposed selection rule behaves in the reported settings. Future work includes designing subset-selection rules that more directly control the residual quantities appearing in the posterior-gap bounds.

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

# A  Appendix Theoretical Analysis

## A.1  Posterior gap for subset-fitted GP under shared hyper-parameters

In this appendix, we analyze the effect of subset-of-data fitting itself. Throughout this section, the full-data GP posterior and the subset-fitted GP posterior are compared under the same kernel hyper-parameters and the same observation-noise variance. This isolates the effect of subset fitting and does not account for the additional error introduced by re-estimating hyper-parameters on the retained subset.

We adopt the standard noisy GP regression model and state the derivation in the zero-mean form; equivalently, the same proof applies after centering the observations with respect to the chosen GP mean function.

Unless otherwise stated, vector norms are Euclidean norms and matrix norms are operator norms.

Theorem 5.1 is the main posterior-gap result used in the remainder of this appendix.

## A.2  Proof of Theorem 5.1

We first introduce the block notation used in the proof. The noisy observations are partitioned according to

$$y_{\mathcal{D}} = \begin{bmatrix} y_{\mathcal{U}} \\ y_{\mathcal{R}} \end{bmatrix}, \qquad k_{\mathcal{D}}(x) = \begin{bmatrix} k_{\mathcal{U}}(x) \\ k_{\mathcal{R}}(x) \end{bmatrix}.$$

The full observation covariance matrix is

$$K_{\mathcal{D}\mathcal{D}} + \sigma_n^2 I = \begin{bmatrix} A & B \\ B^\top & C \end{bmatrix},$$

where

$$A = K_{\mathcal{U}\mathcal{U}} + \sigma_n^2 I, \qquad B = K_{\mathcal{U}\mathcal{R}}, \qquad C = K_{\mathcal{R}\mathcal{R}} + \sigma_n^2 I.$$

Since $K_{\mathcal{D}\mathcal{D}} \succeq 0$ and $\sigma_n^2 > 0$, we have $A \succ 0$. The Schur complement is

$$S = C - B^\top A^{-1} B.$$

The subset-fitted posterior mean and variance are

$$\mu_{\mathcal{U}}(x) = k_{\mathcal{U}}(x)^\top A^{-1} y_{\mathcal{U}},$$

$$\sigma_{\mathcal{U}}^2(x) = k(x,x) - k_{\mathcal{U}}(x)^\top A^{-1} k_{\mathcal{U}}(x).$$

The full-data posterior mean and variance are

$$\mu_{\mathcal{D}}(x) = k_{\mathcal{D}}(x)^\top \begin{bmatrix} A & B \\ B^\top & C \end{bmatrix}^{-1} y_{\mathcal{D}},$$

$$\sigma_{\mathcal{D}}^2(x) = k(x,x) - k_{\mathcal{D}}(x)^\top \begin{bmatrix} A & B \\ B^\top & C \end{bmatrix}^{-1} k_{\mathcal{D}}(x).$$

By the block inverse formula,

$$\begin{bmatrix} A & B \\ B^\top & C \end{bmatrix}^{-1} = \begin{bmatrix} A^{-1} + A^{-1} B S^{-1} B^\top A^{-1} & -A^{-1} B S^{-1} \\ -S^{-1} B^\top A^{-1} & S^{-1} \end{bmatrix}.$$

We now derive the posterior mean gap. Expanding $\mu_{\mathcal{D}}(x)$,

$$\begin{aligned} \mu_{\mathcal{D}}(x) &= \begin{bmatrix} k_{\mathcal{U}}(x) \\ k_{\mathcal{R}}(x) \end{bmatrix}^\top \begin{bmatrix} A^{-1} + A^{-1} B S^{-1} B^\top A^{-1} & -A^{-1} B S^{-1} \\ -S^{-1} B^\top A^{-1} & S^{-1} \end{bmatrix} \begin{bmatrix} y_{\mathcal{U}} \\ y_{\mathcal{R}} \end{bmatrix} \\ &= k_{\mathcal{U}}(x)^\top A^{-1} y_{\mathcal{U}} + \left(k_{\mathcal{R}}(x) - B^\top A^{-1} k_{\mathcal{U}}(x)\right)^\top S^{-1} \left(y_{\mathcal{R}} - B^\top A^{-1} y_{\mathcal{U}}\right). \end{aligned}$$

Define
$$r_{\mathcal{U}} = y_{\mathcal{R}} - B^\top A^{-1} y_{\mathcal{U}}, \qquad c_{\mathcal{U}}(x) = k_{\mathcal{R}}(x) - B^\top A^{-1} k_{\mathcal{U}}(x).$$
Since $\mu_{\mathcal{U}}(x) = k_{\mathcal{U}}(x)^\top A^{-1} y_{\mathcal{U}}$, we obtain
$$\mu_{\mathcal{D}}(x) - \mu_{\mathcal{U}}(x) = c_{\mathcal{U}}(x)^\top S^{-1} r_{\mathcal{U}}.$$

Next, we derive the posterior variance gap. Expanding the quadratic form inside $\sigma_{\mathcal{D}}^2(x)$,

$$k_{\mathcal{D}}(x)^\top \begin{bmatrix} A & B \\ B^\top & C \end{bmatrix}^{-1} k_{\mathcal{D}}(x)$$
$$= k_{\mathcal{U}}(x)^\top A^{-1} k_{\mathcal{U}}(x) + \left( k_{\mathcal{R}}(x) - B^\top A^{-1} k_{\mathcal{U}}(x) \right)^\top S^{-1} \left( k_{\mathcal{R}}(x) - B^\top A^{-1} k_{\mathcal{U}}(x) \right)$$
$$= k_{\mathcal{U}}(x)^\top A^{-1} k_{\mathcal{U}}(x) + c_{\mathcal{U}}(x)^\top S^{-1} c_{\mathcal{U}}(x).$$

Therefore,
$$\sigma_{\mathcal{D}}^2(x) = k(x,x) - k_{\mathcal{U}}(x)^\top A^{-1} k_{\mathcal{U}}(x) - c_{\mathcal{U}}(x)^\top S^{-1} c_{\mathcal{U}}(x),$$
and hence
$$\sigma_{\mathcal{U}}^2(x) - \sigma_{\mathcal{D}}^2(x) = c_{\mathcal{U}}(x)^\top S^{-1} c_{\mathcal{U}}(x).$$

Since $S \succ 0$, the right-hand side is nonnegative, so
$$\sigma_{\mathcal{U}}^2(x) \geq \sigma_{\mathcal{D}}^2(x) \geq 0.$$

We next justify the coarse bound on $\|S^{-1}\|$. Observe that

$$S = \sigma_n^2 I + \left( K_{\mathcal{RR}} - K_{\mathcal{RU}} \left( K_{\mathcal{UU}} + \sigma_n^2 I \right)^{-1} K_{\mathcal{UR}} \right).$$

The term in parentheses is the conditional covariance of the latent function values on $\mathcal{R}$ given noisy observations on $\mathcal{U}$, hence it is positive semidefinite. Therefore,
$$S \succeq \sigma_n^2 I \qquad \implies \qquad \|S^{-1}\| \leq \sigma_n^{-2}.$$

Finally, using
$$|u^\top M v| \leq \|u\| \, \|M\| \, \|v\|,$$
we obtain
$$|\mu_{\mathcal{D}}(x) - \mu_{\mathcal{U}}(x)| \leq \|c_{\mathcal{U}}(x)\| \, \|S^{-1}\| \, \|r_{\mathcal{U}}\|,$$
and
$$0 \leq \sigma_{\mathcal{U}}^2(x) - \sigma_{\mathcal{D}}^2(x) \leq \|c_{\mathcal{U}}(x)\|^2 \, \|S^{-1}\|.$$
Substituting $\|S^{-1}\| \leq \sigma_n^{-2}$ yields the coarse bounds stated in the theorem. $\qquad \square$

### A.3 Proof of Proposition 5.3

Let $z_{\mathcal{D}}$ denote the response vector centered by the chosen GP mean function. Under the zero-mean notation of Theorem 5.1, $z_{\mathcal{D}} = y_{\mathcal{D}}$. Define

$$\lambda := \left( K_{\mathcal{DD}} + \sigma_n^2 I \right)^{-1} z_{\mathcal{D}} = \begin{bmatrix} \lambda_{\mathcal{U}} \\ \lambda_{\mathcal{R}} \end{bmatrix}.$$

The block linear system gives
$$A \lambda_{\mathcal{U}} + B \lambda_{\mathcal{R}} = z_{\mathcal{U}}, \qquad B^\top \lambda_{\mathcal{U}} + C \lambda_{\mathcal{R}} = z_{\mathcal{R}}.$$
From the first equation,
$$\lambda_{\mathcal{U}} = A^{-1} (z_{\mathcal{U}} - B \lambda_{\mathcal{R}}).$$

Substituting this into the second equation yields

$$\left(C - B^\top A^{-1} B\right)\lambda_\mathcal{R} = z_\mathcal{R} - B^\top A^{-1} z_\mathcal{U}.$$

Using $S = C - B^\top A^{-1} B$ and the definition of $r_\mathcal{U}$, this is

$$S\lambda_\mathcal{R} = r_\mathcal{U}, \qquad \lambda_\mathcal{R} = S^{-1} r_\mathcal{U}.$$

The full-data scalar response-gradient score vector is $s = -\lambda$, and therefore

$$s_\mathcal{R} = -S^{-1} r_\mathcal{U}, \qquad r_\mathcal{U} = -S s_\mathcal{R}.$$

Substituting $S^{-1} r_\mathcal{U} = -s_\mathcal{R}$ into Theorem 5.1 gives

$$\mu_\mathcal{D}(x) - \mu_\mathcal{U}(x) = c_\mathcal{U}(x)^\top S^{-1} r_\mathcal{U} = -c_\mathcal{U}(x)^\top s_\mathcal{R}.$$

The pointwise and uniform norm bounds follow from the Cauchy-Schwarz inequality and the definition of $\rho_\mathcal{U}$. $\qquad \square$

## A.4 Uniform posterior distortion bound

**Corollary A.1** (Uniform posterior distortion bound for a fixed retained subset). *Assume $\mathcal{X}$ is compact and the kernel is continuous, so that*

$$\rho_\mathcal{U} := \sup_{x \in \mathcal{X}} \|c_\mathcal{U}(x)\|$$

*is finite. Define*

$$\xi_\mathcal{U} := \|r_\mathcal{U}\|.$$

*Then*

$$\sup_{x \in \mathcal{X}} |\mu_\mathcal{D}(x) - \mu_\mathcal{U}(x)| \le \rho_\mathcal{U}\, \xi_\mathcal{U}\, \|S^{-1}\|,$$

*and*

$$\sup_{x \in \mathcal{X}} \left(\sigma_\mathcal{U}^2(x) - \sigma_\mathcal{D}^2(x)\right) \le \rho_\mathcal{U}^2\, \|S^{-1}\|.$$

*In particular,*

$$\sup_{x \in \mathcal{X}} |\mu_\mathcal{D}(x) - \mu_\mathcal{U}(x)| \le \frac{\rho_\mathcal{U}\, \xi_\mathcal{U}}{\sigma_n^2}, \qquad \sup_{x \in \mathcal{X}} \left(\sigma_\mathcal{U}^2(x) - \sigma_\mathcal{D}^2(x)\right) \le \frac{\rho_\mathcal{U}^2}{\sigma_n^2}.$$

## A.5 A standard-deviation gap lemma

**Lemma A.2** (From variance gap to standard-deviation gap). *For every test point $x$,*

$$|\sigma_\mathcal{U}(x) - \sigma_\mathcal{D}(x)| \le \sqrt{\sigma_\mathcal{U}^2(x) - \sigma_\mathcal{D}^2(x)}.$$

*Consequently,*

$$|\sigma_\mathcal{U}(x) - \sigma_\mathcal{D}(x)| \le \|c_\mathcal{U}(x)\| \sqrt{\|S^{-1}\|} \le \frac{\|c_\mathcal{U}(x)\|}{\sigma_n}.$$

*Proof.* By Theorem 5.1, $\sigma_\mathcal{U}^2(x) \ge \sigma_\mathcal{D}^2(x) \ge 0$. Hence

$$\left(\sigma_\mathcal{U}(x) - \sigma_\mathcal{D}(x)\right)^2 = \sigma_\mathcal{U}^2(x) + \sigma_\mathcal{D}^2(x) - 2\sigma_\mathcal{U}(x)\sigma_\mathcal{D}(x) \le \sigma_\mathcal{U}^2(x) - \sigma_\mathcal{D}^2(x),$$

because

$$\sigma_\mathcal{U}^2(x) - \sigma_\mathcal{D}^2(x) - \left(\sigma_\mathcal{U}(x) - \sigma_\mathcal{D}(x)\right)^2 = 2\sigma_\mathcal{D}(x)\left(\sigma_\mathcal{U}(x) - \sigma_\mathcal{D}(x)\right) \ge 0.$$

Taking square roots yields the first inequality. The remaining bounds follow from Theorem 5.1. $\qquad \square$

### A.6 Proof of Corollary 5.4

For any $x \in \mathcal{X}$, the triangle inequality gives

$$|\alpha_{\mathcal{D}}^{\mathrm{UCB}}(x) - \alpha_{\mathcal{U}}^{\mathrm{UCB}}(x)| \leq |\mu_{\mathcal{D}}(x) - \mu_{\mathcal{U}}(x)| + \sqrt{\beta_t}|\sigma_{\mathcal{D}}(x) - \sigma_{\mathcal{U}}(x)|.$$

Proposition 5.3 bounds the posterior-mean gap by

$$|\mu_{\mathcal{D}}(x) - \mu_{\mathcal{U}}(x)| \leq \|c_{\mathcal{U}}(x)\| \, \|s_{\mathcal{R}}\|.$$

Lemma A.2 bounds the standard-deviation gap by

$$|\sigma_{\mathcal{D}}(x) - \sigma_{\mathcal{U}}(x)| \leq \|c_{\mathcal{U}}(x)\| \sqrt{\|S^{-1}\|}.$$

Combining these two inequalities proves the pointwise bound. Taking the supremum over $x \in \mathcal{X}$ proves the uniform bound. $\qquad\square$

### A.7 Scope and limitations

The results in this appendix should be read with the following scope in mind.

- The posterior-gap theorem is a statement about subset-of-data fitting under shared hyper-parameters and shared observation-noise variance. It isolates the effect of data pruning itself.

- The theorem does not account for additional error introduced by re-estimating hyper-parameters on the retained subset.

- The posterior-gap theorem applies to any retained subset and therefore serves as a diagnostic result for the subset selected by GSSBO; it is not, by itself, an unconditional proof that the GSSBO selection rule always yields a small posterior distortion.

- The residual bridge explains the posterior-mean term for a fixed retained subset. It does not prove that the selection rule minimizes $\|s_{\mathcal{R}}\|$, and it does not control the variance or UCB exploration term by itself.

- The UCB acquisition-stability corollary is not a cumulative-regret bound and does not prove that the selection rule always produces small acquisition distortion.

## B Background: Gaussian Process Regression and Bayesian Optimization

**GP regression.** A Gaussian process prior $f \sim \mathcal{GP}(\mu(\cdot), k(\cdot, \cdot; \theta))$ states that any finite collection of function values is jointly Gaussian. Given observations $\mathbf{y} = (y_1, \ldots, y_n)^\top$ at inputs $\mathbf{X} = (x_1, \ldots, x_n)$ with $y_i = f(x_i) + \varepsilon_i$, $\varepsilon_i \sim \mathcal{N}(0, \sigma_n^2)$, let $\mathbf{K}$ be the kernel Gram matrix and $\mathbf{K}_y = \mathbf{K} + \sigma_n^2 I$. Writing $\mathbf{z} = \mathbf{y} - \boldsymbol{\mu}$, the posterior at a test point $x$ is Gaussian with

$$\mu_n(x) = \mu(x) + k_n(x)^\top \mathbf{K}_y^{-1} \mathbf{z}, \qquad \sigma_n^2(x) = k(x, x) - k_n(x)^\top \mathbf{K}_y^{-1} k_n(x),$$

where $k_n(x) = (k(x_1, x), \ldots, k(x_n, x))^\top$. Kernel hyperparameters $\theta$ and $\sigma_n^2$ are typically estimated by maximizing the log marginal likelihood in Eq. 2. Factorizing $\mathbf{K}_y$ costs $\mathcal{O}(n^3)$, which is the scaling bottleneck addressed in this paper.

**Bayesian optimization.** BO seeks $x^\star = \arg\max_{x \in \mathcal{X}} f(x)$ for an expensive black-box $f$. Starting from an initial design, BO iterates: (i) fit the GP posterior to the current data; (ii) maximize an acquisition function $\alpha(x; p_t(f))$ that trades off exploitation and exploration to pick $x_t$; (iii) evaluate $y_t = f(x_t)$ and append $(x_t, y_t)$ to the dataset. Common acquisition functions include the upper confidence bound, $\alpha^{\mathrm{UCB}}(x) = \mu_n(x) + \sqrt{\beta_t}\, \sigma_n(x)$, expected improvement, $\alpha^{\mathrm{EI}}(x) = \mathbb{E}[\max(f(x) - f^+, 0)]$ with incumbent $f^+$, and Thompson sampling, which maximizes a posterior sample. This paper uses UCB; the subset-maintenance mechanism is agnostic to this choice because it only changes which observations the GP is refit on.

## C   Appendix Experiments

### C.1   Additional Implementation Details

Fixing $M$ (or the switch iteration) in advance removes the wall-clock dependence introduced by the $Z$-trigger and makes runs deterministic given the seeds; the fixed-$M$ results in Appendix C.5 use this mode.

**Greedy cosine-diversity selection.**   For each selection pool $\mathcal{P}_t$, we compute the pool-specific sensitivity embeddings $\mathbf{g}_i^{(\mathcal{P}_t)}$ and normalize them as $\mathbf{v}_i = \mathbf{g}_i^{(\mathcal{P}_t)}/\|\mathbf{g}_i^{(\mathcal{P}_t)}\|_2$. ~~The latest observation is inserted first. We then greedily choose the remaining $M-1$ samples by selecting candidates with small cumulative cosine similarity to the already-retained set.~~ The initial design and the latest observation are inserted first. We then greedily choose the remaining $M - n_0 - 1$ samples by selecting candidates with small cumulative cosine similarity to the already retained set. After the pairwise pool scores $C_{ij} = \mathbf{v}_i^\top \mathbf{v}_j$ are available, we maintain $h_i = \sum_{j\in\mathcal{S}} C_{ij}$ for each unselected candidate $i$, where $\mathcal{S}$ is the current retained set. When candidate $a$ is added, the scores are updated as $h_i \leftarrow h_i + C_{ia}$. Thus the greedy stage costs $\mathcal{O}(q_t M)$ after the pool-level cosine scores have been constructed or cached; embedding construction and cosine-score construction are counted separately in the overhead terms described in Section 4.3.

**Runtime-triggered buffer initialization.**   The parameter $Z$ is used only to initialize the buffer size at the first switch. In the reported experiments, GSSBO and RSSBO use the same runtime-triggered rule, hardware setting, and evaluation protocol; after the switch, $M$ is kept fixed. Since the trigger depends on measured iteration time, the exact switching iteration may vary across machines. For deterministic reruns, one can instead fix $M$ directly or fix the switching iteration and then apply the same subset-maintenance rule. This affects only the practical initialization of $M$, not the cosine-diversity selection rule once $M$ and $\mathcal{P}_t$ are fixed.

### C.2   Supplementary experiments

Figure 5 and 6 compare the cumulative runtime over 1000 iterations on Eggholder2, Levy20, Rastrigin100, and the diabetes hyperparameter optimization task.

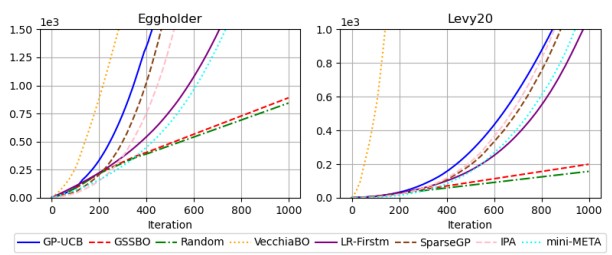

Figure 5: Cumulative runtime (seconds) on Eggholder2 and Levy20.

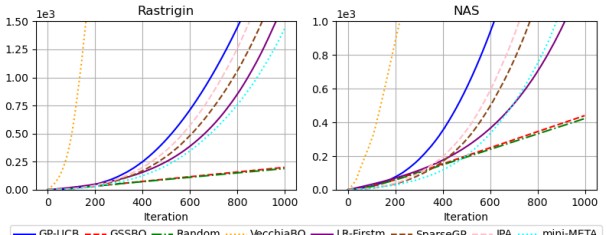

Figure 6: Cumulative runtime (seconds) on Rastrigin100 and the diabetes hyperparameter optimization task.

### C.3   The scalar-score variant: definition and direct comparison

The scalar-score variant replaces the vector embeddings by the scalar scores $s_i$ computed on the same selection pool. For one-dimensional scores, cosine similarity reduces to the sign comparison $\mathrm{sign}(s_i s_j)$; scores with magnitude below $10^{-12}$ are floored at that value for normalization. All other components are unchanged—the same greedy cumulative-cosine minimization, the same forced retention of the latest observation, and the same retained-initial-design default—so the variant differs from the default GSSBO configuration only in the selection representation. Table 2 reports the direct comparison at matched retention.

The vector rule attains lower final cumulative regret than the scalar variant on all six tasks under the dynamic-$Z$ configuration, and the fixed-$M$=100 rows confirm this direction on Eggholder2 and Hart6, at the same subset-GP cost (the variant differs only in the selection representation). We therefore retain

Table 2: Scalar-score variant vs. vector default (final cumulative regret, mean $\pm$ 95% CI over the 50 runs).

| Config | Task | Scalar | Vector | Better |
|---|---|---|---|---|
| dynamic-$Z$ | Eggholder2 | $1919 \pm 228$ | $1425 \pm 121$ | vector |
| dynamic-$Z$ | Hart6 | $5453 \pm 86$ | $3874 \pm 119$ | vector |
| dynamic-$Z$ | Levy20 ($\times 10^5$) | $2.13 \pm 0.11$ | $1.504 \pm 0.071$ | vector |
| dynamic-$Z$ | Powell50 ($\times 10^7$) | $7.19 \pm 0.30$ | $5.479 \pm 0.280$ | vector |
| dynamic-$Z$ | Rastrigin100 ($\times 10^6$) | $1.889 \pm 0.001$ | $1.232 \pm 0.009$ | vector |
| dynamic-$Z$ | Diabetes | $435 \pm 2$ | $89.1 \pm 6.2$ | vector |
| fixed $M{=}100$ | Eggholder2 | $1991 \pm 199$ | $1676 \pm 95$ | vector |
| fixed $M{=}100$ | Hart6 | $5505 \pm 103$ | $3938 \pm 134$ | vector |

the vector rule as the default and report the scalar rule as an empirical variant. This ablation tests the practical representation choice directly; the scalar scores enter Proposition 5.3, but that proposition does not theoretically validate either selection rule.

## C.4 Residual-quantity diagnostics for the selection rule

This experiment measures the quantities appearing in the Theorem 5.1 bounds directly, comparing the vector cosine-diversity rule with random selection under matched conditions. For each task (Eggholder2, Hart6) we generate a shared set of full-data GP-UCB trajectories (600 iterations, $n_0 = 20$); at each checkpoint $t \in \{100, \ldots, 600\}$ the kernel hyperparameters and observation-noise variance are fitted by maximum likelihood on the full $\mathcal{D}_t$ and shared by all posteriors, as in Theorem 5.1. Each rule then builds a size-$M{=}100$ subset from the same $\mathcal{D}_t$ under matched retention: both rules retain the initial design and the latest observation, as in the single-factor grid of Table 5, so the rules differ only in how the remaining slots are filled. Random subsets are redrawn ten times per checkpoint and averaged within seed. Table 3 reports the final checkpoint on standardized responses, with the suprema approximated over a grid of 2000 uniform candidates augmented with the current observations.

Table 3: Residual-quantity diagnostics (consolidated reference implementation; shared full-data GP-UCB trajectories per task; checkpoint $t = 600$; $M = 100$; matched retention of the initial design and the latest observation): mean $\pm$ 95% CI over seeds. The posterior-gap row is on standardized responses; lower is better for every quantity, and bold marks the better rule on the quantities with a significant paired difference (no difference is detected for $\|s_\mathcal{R}\|$ and $\|S^{-1}\|$).

| Task | Quantity | Vector selection | Matched random selection |
|---|---|---|---|
| Eggholder2 | $\|r_\mathcal{U}\|$ | $\mathbf{8.9 \pm 1.7}$ | $13.0 \pm 0.8$ |
| | $\rho_\mathcal{U}$ | $\mathbf{6.8 \pm 1.3}$ | $9.4 \pm 1.4$ |
| | $\|s_\mathcal{R}\|$ | $219 \pm 12$ | $211 \pm 11$ |
| | $\|S^{-1}\|$ | $106 \pm 14$ | $106 \pm 14$ |
| | $\sup_x |\mu_\mathcal{D} - \mu_\mathcal{U}|$ | $\mathbf{3.50 \pm 0.51}$ | $4.59 \pm 0.35$ |
| Hart6 | $\|r_\mathcal{U}\|$ | $\mathbf{3.40 \pm 0.97}$ | $4.26 \pm 0.99$ |
| | $\rho_\mathcal{U}$ | $\mathbf{1.21 \pm 0.27}$ | $1.65 \pm 0.19$ |
| | $\|s_\mathcal{R}\|$ | $561 \pm 50$ | $536 \pm 56$ |
| | $\|S^{-1}\|$ | $1102 \pm 890$ | $1089 \pm 860$ |
| | $\sup_x |\mu_\mathcal{D} - \mu_\mathcal{U}|$ | $\mathbf{1.01 \pm 0.19}$ | $1.34 \pm 0.16$ |

At matched default retention the vector rule attains smaller discarded-response residual $\|r_\mathcal{U}\|$, conditional cross-covariance residual $\rho_\mathcal{U}$, and realized posterior-mean gap than random selection on both tasks (by 23–38%, computed as geometric means of the per-seed paired ratios, vector/random; paired Wilcoxon $p \leq 0.03$), so the Proposition 5.3 product $\rho_\mathcal{U}\|s_\mathcal{R}\|$ is 28–29% smaller under the same paired-ratio measure. $\|s_\mathcal{R}\|$ is slightly larger (4–5%): the rule does not target this response-dependent quantity, and it preferentially discards near-duplicate observations, whose leave-one-out variances are small and whose score entries are therefore

large in magnitude (Section 4.1), so a slightly larger $\|s_{\mathcal{R}}\|$ is consistent with the selection geometry; for $\|S^{-1}\|$ no difference between the rules was detected, consistent with its extreme eigenvalue being governed by near-duplicate pairs that both rules discard. These are empirical diagnostics on two tasks under the stated protocol, not a guarantee that the rule controls the bound quantities in general; Levy20 is excluded as degenerate, since its full-data length-scales make the kernel nearly constant and all conditional residuals trivial.

## C.5 Runtime decomposition and component ablations

Table 6 specifies each compared method's configuration, and Table 7 reports the total runtime of the Figure 3 experiments together with its per-component decomposition, including the residual overhead so that shares sum to 100% up to rounding. Table 8 reports the same quantities for the remaining four tasks (the experiments of Figures 5 and 6).

**Component ablations (reference implementation, matched seeds, dynamic-$Z$ unless noted).** Table 4 varies one component of the configuration at a time. On Hart6, k-means++ in the same embedding space is competitive with greedy cosine, and forcing the newest observation on or off makes little difference; we report both directions as-is, and both effects are within noise on Eggholder2. Fixed buffers move monotonically toward full-GP performance with larger $M$ on both tasks. Max–min kernel-distance selection is not covered by these ablations and is left as future work.

Table 4: Component ablations (reference implementation, matched seeds, dynamic-$Z$ unless noted): final cumulative regret (mean $\pm$ 95% CI), varying one component of the configuration at a time. All rows are run without initial-design retention; that factor is isolated separately in the single-factor grid below.

| Configuration | Eggholder2 | Hart6 |
|---|---|---|
| greedy cosine, force-latest, dynamic-$Z$ | $1823 \pm 89$ | $5191 \pm 102$ |
| k-means++ in the same embedding space | $1914 \pm 89$ | $5415 \pm 231$ |
| without forcing the newest observation | $1982 \pm 95$ | $5388 \pm 336$ |
| fixed $M{=}50$ | $2021 \pm 109$ | $6258 \pm 159$ |
| fixed $M{=}100$ | $1675 \pm 96$ | $5453 \pm 156$ |
| fixed $M{=}200$ | $1597 \pm 82$ | $3989 \pm 186$ |

**Single-factor grid (fixed $M{=}100$, independent RNG streams, 50 matched seeds).** Table 5 isolates the selection rule from initial-design retention, varying one factor at a time under a fixed buffer. Retaining the initial design clearly improves the vector and random rules on both tasks, while its effect on the scalar rule is negligible; at matched retention the vector rule attains the lowest final cumulative regret on both tasks, with non-overlapping 95% confidence intervals against random selection.

Table 5: Single-factor grid (fixed $M{=}100$, independent RNG streams, 50 matched seeds): final cumulative regret (mean $\pm$ 95% CI) for each selection rule with and without initial-design retention.

| Task | Selection rule | without retention | with retention |
|---|---|---|---|
| Eggholder2 | vector | $1801 \pm 74$ | $1676 \pm 95$ |
| Eggholder2 | random | $2603 \pm 73$ | $1977 \pm 68$ |
| Eggholder2 | scalar | $2037 \pm 201$ | $1991 \pm 199$ |
| Hart6 | vector | $5498 \pm 206$ | $3938 \pm 134$ |
| Hart6 | random | $5925 \pm 119$ | $4972 \pm 92$ |
| Hart6 | scalar | $5520 \pm 101$ | $5505 \pm 103$ |

## C.6 Selection-pool and embedding-refresh ablation

We isolate the effect of the selection-pool construction at a fixed buffer size $M = 100$ (matched seeds, 1000 iterations, UCB, per-iteration MLE unless noted). We compare the full-history pool $\mathcal{P}_t = \mathcal{D}_t$ used in the

Table 6: Configurations used for the reference-implementation experiments in this appendix. All methods share the same protocol: UCB with $\beta_t^{1/2} = \sqrt{2}$, uniform candidate sets of adaptive size (10,000 for $d \leq 10$, 5,000 for $d \leq 50$, 2,000 otherwise) drawn per iteration, Latin-hypercube initial designs of 20 points, 1000 iterations, and matched seeds. The next point is selected by exhaustive argmax of the acquisition over the shared candidate set (no gradient-based acquisition optimization). Exact-GP methods and the Nyström baseline are implemented on scikit-learn's `GaussianProcessRegressor`; SparseGP/IPA use `gpytorch`; VecchiaBO uses the official `pyvecch` package.

| Method | Surrogate | Hyperparameter learning | Method-specific settings |
|---|---|---|---|
| GP-UCB | exact GP, ARD Matérn 5/2 | per-iteration MLE (2 restarts) | full data |
| GSSBO (default) | exact GP on retained subset | per-iteration MLE (2 restarts) | vector cosine rule; retains initial design and latest point; dynamic-$Z$ ($Z$=4) |
| GSSBO-scalar | exact GP on retained subset | per-iteration MLE (2 restarts) | sign-balanced scalar rule; same retention and trigger |
| RSSBO | exact GP on retained subset | per-iteration MLE (2 restarts) | random selection; retains latest point only |
| LR-First $m$ | Nyström on first $m$ samples | shared MLE hyperparameters | $m = \max(9, \min(100, \lfloor t/10 \rfloor + 9, n/2))$, growing to $\leq 100$ |
| SparseGP (SVGP) | variational sparse GP (gpytorch) | Adam(0.01), 100 epochs/iter | 40 inducing points, random init, locations learned |
| IPA | variational sparse GP (gpytorch) | Adam(0.01), 100 epochs/iter | 40 inducing points, quality–diversity init, locations learned |
| mini-META | exact GP on distinct points | per-iteration MLE (2 restarts) | repeat evaluation until $\sqrt{\sigma_n^2/\text{count}} < \tau$=0.1 (max 20 repeats) |
| VecchiaBO | Vecchia approximation (official `pyvecch`) | per-iteration refit | nearest-neighbor conditioning sets, size grown with $t$ |

Table 7: Total runtime at iteration 1000 for the experiments of Figure 3 (mean $\pm$ 95% CI over the 50 runs), share of GP-UCB, and per-component decomposition (% of total; Other includes bookkeeping so rows sum to 100 up to rounding).

| Task | Method | Total (s) | % GP-UCB | Refit | Embed | Select | Acq. | Other |
|---|---|---|---|---|---|---|---|---|
| Hart6 | GP-UCB | $2{,}433 \pm 249$ | 100.0% | 98.7 | 0.0 | 0.0 | 1.3 | 0.0 |
| Hart6 | GSSBO | $211 \pm 23$ | 8.7% | 79.2 | 9.0 | 0.1 | 7.6 | 4.1 |
| Hart6 | IPA | $3{,}172 \pm 207$ | 130.4% | 98.4 | 0.0 | 0.0 | 1.6 | 0.0 |
| Hart6 | LR-First $m$ | $739 \pm 58$ | 30.4% | 89.9 | 0.0 | 0.0 | 10.0 | 0.0 |
| Hart6 | mini-META | $881 \pm 44$ | 36.2% | 96.9 | 0.0 | 0.0 | 3.1 | 0.0 |
| Hart6 | RSSBO | $209 \pm 22$ | 8.6% | 92.3 | 0.0 | 0.1 | 7.7 | 0.0 |
| Hart6 | SparseGP | $1{,}577 \pm 69$ | 64.8% | 98.5 | 0.0 | 0.0 | 1.5 | 0.0 |
| Hart6 | VecchiaBO | $9{,}073 \pm 996$ | 372.9% | 94.9 | 0.0 | 0.0 | 5.1 | 0.0 |
| Powell50 | GP-UCB | $2{,}495 \pm 108$ | 100.0% | 98.4 | 0.0 | 0.0 | 1.6 | 0.0 |
| Powell50 | GSSBO | $236 \pm 14$ | 9.5% | 56.7 | 21.1 | 0.3 | 14.7 | 7.3 |
| Powell50 | IPA | $2{,}149 \pm 98$ | 86.1% | 99.2 | 0.0 | 0.0 | 0.8 | 0.0 |
| Powell50 | LR-First $m$ | $1{,}659 \pm 90$ | 66.5% | 83.6 | 0.0 | 0.0 | 16.3 | 0.1 |
| Powell50 | mini-META | $1{,}804 \pm 44$ | 72.3% | 97.0 | 0.0 | 0.0 | 3.0 | 0.0 |
| Powell50 | RSSBO | $218 \pm 34$ | 8.7% | 83.6 | 0.0 | 0.1 | 16.2 | 0.1 |
| Powell50 | SparseGP | $2{,}061 \pm 36$ | 82.6% | 99.2 | 0.0 | 0.0 | 0.8 | 0.0 |
| Powell50 | VecchiaBO | $13{,}785 \pm 864$ | 552.5% | 95.9 | 0.0 | 0.0 | 4.1 | 0.0 |

headline experiments against sliding-window pools of the most recent $|\mathcal{W}_t| \in \{M, 2M, 4M\}$ observations, and, under hyperparameters frozen at the buffer switch, recomputing the embeddings each step versus an exact cached rank-one refresh. Table 9 reports final cumulative regret, total runtime, and the embedding-construction share.

Three observations. First, restricting the pool sharply reduces the embedding overhead (Hart6: 29.0s to 0.5s) while leaving total runtime dominated by the subset-GP refit. Second, the reported efficiency gain is not an artifact of a favorable pool choice: the headline configuration uses the most expensive full-history pool with its cost included, and the sliding-window pools match or improve its regret at lower embedding cost, so pool restriction is an additional optimization opportunity rather than a requirement of the reported results. Third, the cached rank-one refresh is exact under frozen hyperparameters—identical regret to recomputation—while further reducing the embedding cost. The regret levels in this table correspond to the fixed-$M$ configuration and are therefore not directly comparable to the dynamic-$Z$ headline configuration.

Table 8: Total runtime at iteration 1000 for the experiments of Figures 5 and 6 (mean $\pm$ 95% CI over the 50 runs), share of GP-UCB, and per-component decomposition (% of total; Other includes bookkeeping so rows sum to 100 up to rounding).

| Task | Method | Total (s) | % GP-UCB | Refit | Embed | Select | Acq. | Other |
|---|---|---|---|---|---|---|---|---|
| Eggholder2 | GP-UCB | $9{,}578 \pm 542$ | 100.0% | 95.9 | 0.0 | 0.0 | 4.1 | 0.0 |
| Eggholder2 | GSSBO | $894 \pm 66$ | 9.3% | 83.7 | 4.9 | 0.2 | 9.1 | 2.1 |
| Eggholder2 | IPA | $9{,}871 \pm 119$ | 103.1% | 98.5 | 0.0 | 0.0 | 1.4 | 0.0 |
| Eggholder2 | LR-First $m$ | $3{,}247 \pm 212$ | 33.9% | 88.5 | 0.0 | 0.0 | 11.4 | 0.0 |
| Eggholder2 | mini-META | $3{,}160 \pm 116$ | 33.0% | 94.1 | 0.0 | 0.0 | 5.9 | 0.0 |
| Eggholder2 | RSSBO | $847 \pm 76$ | 8.8% | 90.9 | 0.0 | 0.0 | 9.1 | 0.0 |
| Eggholder2 | SparseGP | $12{,}190 \pm 724$ | 127.3% | 98.6 | 0.0 | 0.0 | 1.4 | 0.0 |
| Eggholder2 | VecchiaBO | $12{,}738 \pm 890$ | 133.0% | 94.9 | 0.0 | 0.0 | 5.1 | 0.0 |
| Levy20 | GP-UCB | $1{,}493 \pm 87$ | 100.0% | 97.4 | 0.0 | 0.0 | 2.6 | 0.0 |
| Levy20 | GSSBO | $206 \pm 26$ | 13.8% | 63.7 | 15.4 | 0.4 | 14.2 | 6.3 |
| Levy20 | IPA | $1{,}533 \pm 128$ | 102.7% | 98.2 | 0.0 | 0.0 | 1.8 | 0.0 |
| Levy20 | LR-First $m$ | $1{,}089 \pm 83$ | 72.9% | 78.5 | 0.0 | 0.0 | 21.3 | 0.1 |
| Levy20 | mini-META | $1{,}112 \pm 121$ | 74.5% | 94.5 | 0.0 | 0.0 | 5.5 | 0.0 |
| Levy20 | RSSBO | $161 \pm 21$ | 10.8% | 80.8 | 0.0 | 0.1 | 18.9 | 0.1 |
| Levy20 | SparseGP | $1{,}479 \pm 114$ | 99.1% | 98.2 | 0.0 | 0.0 | 1.8 | 0.0 |
| Levy20 | VecchiaBO | $39{,}293 \pm 1{,}507$ | 2631.8% | 96.7 | 0.0 | 0.0 | 3.3 | 0.0 |
| Rastrigin100 | GP-UCB | $2{,}536 \pm 135$ | 100.0% | 99.6 | 0.0 | 0.0 | 0.4 | 0.0 |
| Rastrigin100 | GSSBO | $207 \pm 19$ | 8.2% | 67.9 | 21.0 | 0.2 | 4.5 | 6.4 |
| Rastrigin100 | IPA | $2{,}335 \pm 107$ | 92.1% | 99.0 | 0.0 | 0.0 | 0.9 | 0.0 |
| Rastrigin100 | LR-First $m$ | $1{,}685 \pm 136$ | 66.4% | 95.2 | 0.0 | 0.0 | 4.7 | 0.1 |
| Rastrigin100 | mini-META | $1{,}416 \pm 157$ | 55.8% | 99.3 | 0.0 | 0.0 | 0.7 | 0.0 |
| Rastrigin100 | RSSBO | $197 \pm 32$ | 7.8% | 95.0 | 0.0 | 0.1 | 4.9 | 0.1 |
| Rastrigin100 | SparseGP | $1{,}933 \pm 103$ | 76.2% | 99.0 | 0.0 | 0.0 | 0.9 | 0.0 |
| Rastrigin100 | VecchiaBO | $74{,}862 \pm 955$ | 2952.0% | 97.9 | 0.0 | 0.0 | 2.1 | 0.0 |
| Diabetes | GP-UCB | $3{,}262 \pm 179$ | 100.0% | 96.5 | 0.0 | 0.0 | 3.2 | 0.3 |
| Diabetes | GSSBO | $442 \pm 15$ | 13.5% | 60.0 | 14.1 | 0.1 | 10.4 | 15.3 |
| Diabetes | IPA | $2{,}562 \pm 81$ | 78.5% | 93.1 | 0.0 | 0.0 | 1.9 | 5.0 |
| Diabetes | LR-First $m$ | $1{,}312 \pm 50$ | 40.2% | 66.1 | 0.0 | 0.0 | 15.0 | 18.9 |
| Diabetes | mini-META | $1{,}453 \pm 91$ | 44.5% | 96.5 | 0.0 | 0.0 | 3.2 | 0.3 |
| Diabetes | RSSBO | $426 \pm 18$ | 13.1% | 74.6 | 0.0 | 0.0 | 14.9 | 10.5 |
| Diabetes | SparseGP | $2{,}179 \pm 96$ | 66.8% | 92.9 | 0.0 | 0.0 | 2.0 | 5.1 |
| Diabetes | VecchiaBO | $15{,}184 \pm 695$ | 465.5% | 90.3 | 0.0 | 0.0 | 9.2 | 0.5 |

Table 9: Selection-pool and embedding-refresh ablation (mean $\pm$ 95% CI over matched seeds; times are per-run totals in seconds).

| Task | Pool / refresh | Final cum. regret | Total time (s) | Embedding time (s) |
|---|---|---|---|---|
| Hart6 | $\mathcal{P}_t = \mathcal{D}_t$ | $5656 \pm 46$ | 466 | 29.0 |
| Hart6 | window $4M$ | $5328 \pm 39$ | 455 | 8.5 |
| Hart6 | window $2M$ | $5359 \pm 64$ | 429 | 1.9 |
| Hart6 | window $M$ | $2708 \pm 72$ | 411 | 0.5 |
| Hart6 | $\mathcal{D}_t$, frozen, recompute | $5948 \pm 894$ | 96 | 27.7 |
| Hart6 | $\mathcal{D}_t$, frozen, cached | $5948 \pm 894$ | 91 | 17.7 |
| Powell50 | $\mathcal{P}_t = \mathcal{D}_t$ | $(7.57 \pm 0.72) \times 10^7$ | 292 | 38.6 |
| Powell50 | window $4M$ | $(7.44 \pm 0.38) \times 10^7$ | 275 | 9.9 |
| Powell50 | window $2M$ | $(7.32 \pm 0.17) \times 10^7$ | 269 | 2.6 |
| Powell50 | window $M$ | $(6.94 \pm 0.17) \times 10^7$ | 287 | 0.9 |
| Powell50 | $\mathcal{D}_t$, frozen, recompute | $(7.23 \pm 0.96) \times 10^7$ | 99 | 38.8 |
| Powell50 | $\mathcal{D}_t$, frozen, cached | $(7.23 \pm 0.96) \times 10^7$ | 80 | 18.2 |

## C.7 Hyperparameter re-estimation ablation

Complementing the remark in Section 5, we compare GSSBO with kernel hyperparameters re-estimated by maximum likelihood at every iteration against hyperparameters frozen at the buffer switch (dynamic-$Z$ default configuration, 1000 iterations); full-data GP-UCB with per-iteration re-estimation is included as a reference.

Freezing at the switch increases final cumulative regret moderately on both tasks (by 8–15% here) while removing the per-iteration hyperparameter optimization, so it is a reasonable trade-off when the refitting

Table 10: Hyperparameter re-estimation ablation (mean $\pm$ 95% CI over the 50 runs).

| Task | Setting | Final cum. regret |
|---|---|---|
| Eggholder2 | GSSBO, per-iteration MLE | $1425 \pm 121$ |
| Eggholder2 | GSSBO, frozen at switch | $1642 \pm 144$ |
| Eggholder2 | full-data GP-UCB, per-iteration MLE | $1196 \pm 109$ |
| Hart6 | GSSBO, per-iteration MLE | $3874 \pm 119$ |
| Hart6 | GSSBO, frozen at switch | $4184 \pm 169$ |
| Hart6 | full-data GP-UCB, per-iteration MLE | $3540 \pm 105$ |

budget is constrained (and it enables the exact cached embedding refresh of the pool ablation); per-iteration re-estimation remains the default. This quantifies the re-estimation effect that the fixed-hyperparameter theory deliberately excludes, and we scope the theoretical claims accordingly.

## C.8   Subset Samples Distribution Study

Figure 7 illustrates the sample distribution of GSSBO and standard GP-UCB, on the first two dimensions of the Hartmann6 function. In this experiment, we recorded the first 200 sequential samples from a standard BO process and constrained the buffer size to 100. The objective is to identify the global minimum, and darker-colored samples correspond to values closer to the optimal. During the optimization process, only a small number of samples are located near the optimal value. In this run, GSSBO selection retains more samples near low-value regions while still covering diverse observed locations, as indicated by a higher number of darker-colored samples in the middle panel. In contrast, the random selection strategy appears to retain fewer samples near the optimal or suboptimal regions in this run, as represented by the retention of many lighter-colored samples in the right panel.

While BO algorithms are designed to balance exploitation and exploration, with limited budget in practice, they can concentrate evaluations around current best regions before shifting to exploration (Wang and Ng, 2020), which may affect optimization performance. With GSSBO selection, the relative density of retained samples near low-value regions increases in this visualization. This pattern may help avoid excessive concentration around the current best region and encourage broader coverage of the observed search space.

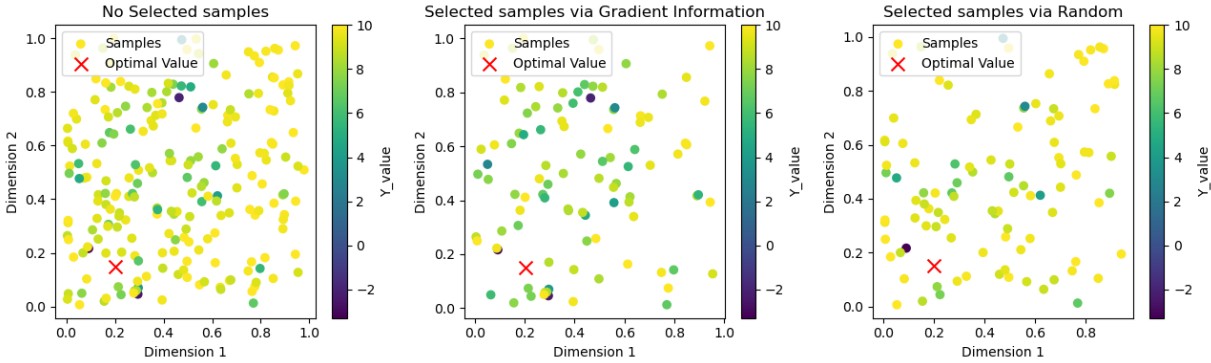

Figure 7: Sample distribution: GP-UCB (left), GSSBO (middle), and RSSBO (right).

## C.9   Experiments on K-means++ Selection

Oglic and Gärtner (2017) and Hayakawa et al. (2023) proposed to select a subset in RKHS, then employed them to construct the Nyström low-rank approximation. We included it as an additional baseline. In these experiments, Fig. 8 shows that K-means++ has no clear cumulative-regret advantage over GSSBO and incurs higher runtime.

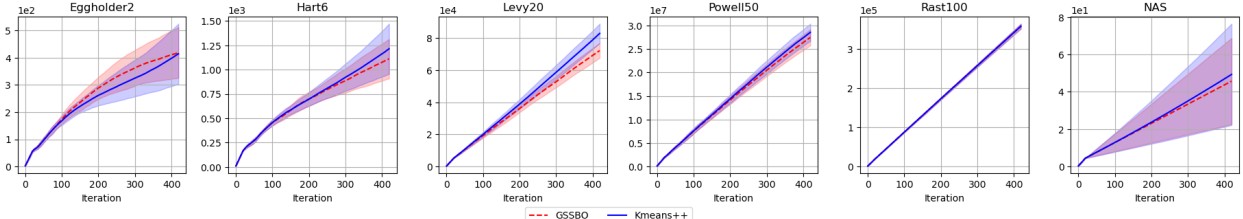

Figure 8: Cumulative regret of GSSBO and K-means++ on Eggholder2, Hart6, Levy20, Powell50, Rastrigin100, and the diabetes hyperparameter optimization task.

## C.10 Experiments on high-dimensional BO methods

We noticed that several compared methods show degraded performance on high-dimensional tasks, so we included additional high-dimensional baselines: REMBO (Wang et al., 2016) and HeSBO (Nayebi et al., 2019). The results are shown in Fig. 9. The difference between the three methods is subtle on Levy20, while REMBO and HeSBO perform worse on Powell50 and Rastrigin100 in our results, possibly because the random embedding can discard task-relevant directions.

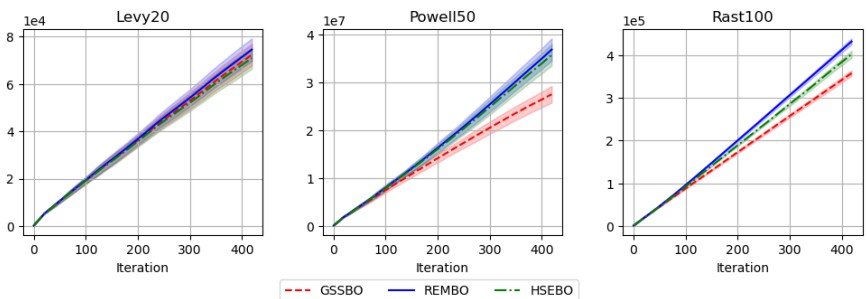

Figure 9: Cumulative regret of GSSBO, REMBO, and HeSBO on the Levy20, Powell50, and Rastrigin100 functions.

## C.11 Experiments on other surrogates

Although our theoretical analysis focuses on Gaussian Process surrogates, we also explore whether the same subset-maintenance idea can be paired empirically with other surrogate models. For Bayesian neural networks (BNN) (Mullachery et al., 2018) and Deep Kernel (DK) methods (Wilson et al., 2016), the selection step uses surrogate-specific selection features rather than the GP-specific precision-column form in Eq. 3. The corresponding results are shown in Fig. 10.

## C.12 Auxiliary RMSE Diagnostic for Subset GP Fits

As an auxiliary posterior-mean diagnostic, we compare the RMSE of three surrogate models on a test dataset: (i) the full-data GP used in GP-UCB, (ii) the GP trained on the GSSBO-selected subset, and (iii) the GP trained on a randomly chosen subset of the same size (RSSBO). Subset selection starts at the 60th iteration. At each iteration, we construct the corresponding GP surrogate and compute the root mean square error (RMSE) on a test set of size $n_{\text{test}}$, where $\hat{\mu}_t(\tilde{x}_j)$ denotes the posterior mean prediction of the surrogate at the $j$-th test input $\tilde{x}_j$:

$$\text{RMSE}_t = \sqrt{\frac{1}{n_{\text{test}}} \sum_{j=1}^{n_{\text{test}}} \left(\hat{\mu}_t(\tilde{x}_j) - f(\tilde{x}_j)\right)^2}.$$

As shown in Figure 11, GSSBO often attains RMSE values close to the full-data GP in this experiment, while RSSBO shows higher error and variance in these runs. This suggests that the GSSBO-selected subset can

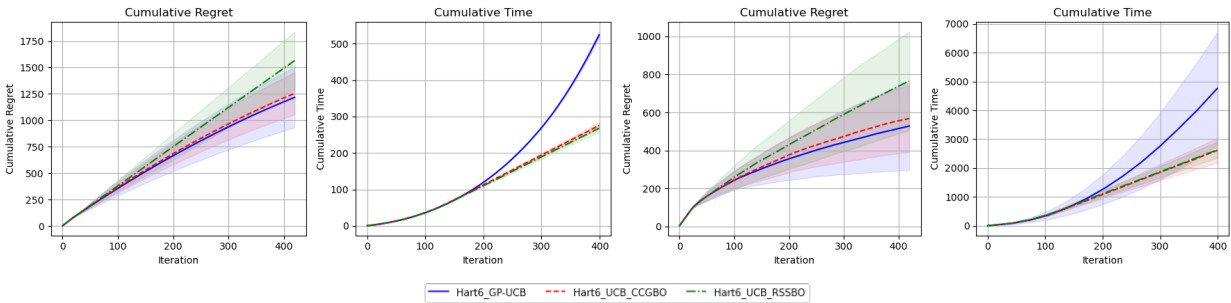

Figure 10: Surrogate-specific Hart6 experiments using Deep Kernel (left pair) and BNN (right pair) surrogate settings. Within each pair, the first panel reports cumulative regret and the second reports cumulative runtime.

preserve posterior-mean prediction accuracy better than a random subset of the same size in this diagnostic. This result should be interpreted as supplementary evidence only: RMSE measures posterior-mean prediction error and does not directly test posterior variance, UCB acquisition values, or the residual quantities appearing in the theoretical posterior-gap bounds.

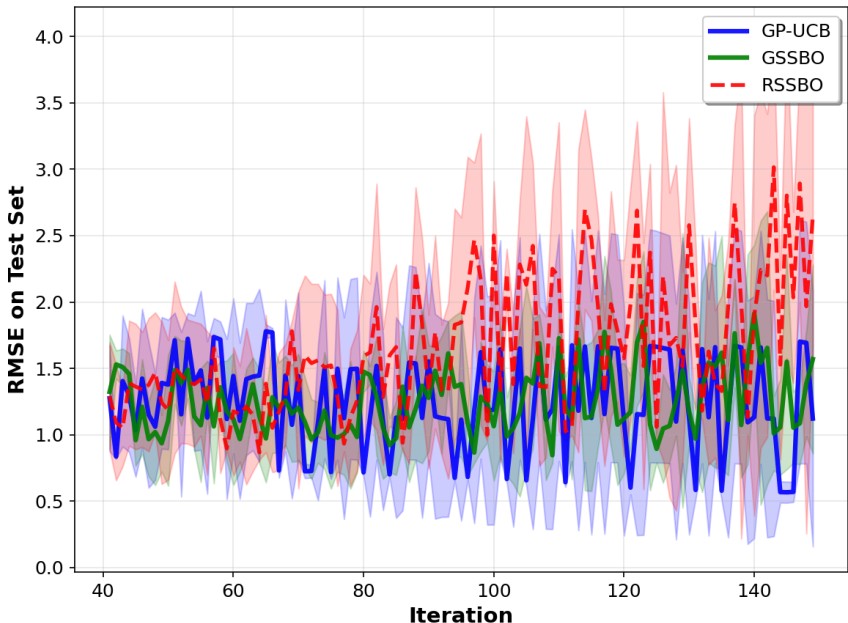

Figure 11: Auxiliary RMSE diagnostic over iterations for full-data, GSSBO subset, and RSSBO subset GP fits.

