# OpenReview forum: "Gradient-based Sample Selection for Faster Bayesian Optimization"
_TMLR — Under review for TMLR_

### Review · Reviewer_TGL6 · 2026-06-17

**Summary Of Contributions:**

This paper proposes Gradient-based Sample Selection Bayesian Optimization (GSSBO), a subset-maintenance method for accelerating GP-based Bayesian optimization. The method retains a fixed-size subset of observations using GP response-sensitivity embeddings and a cosine-diversity rule, and then refits the GP surrogate only on this retained subset.
The paper contributes an algorithmic framework in Section 4, a conditional posterior-gap analysis for fixed retained subsets in Section 5, and empirical evaluations on synthetic benchmarks and a diabetes hyperparameter-optimization task in Section 6.

**Additional Comments:**

None

**Audience:**

Yes

**Audience Explanation:**

The paper addresses a relevant problem for TMLR readers interested in Bayesian optimization, scalable Gaussian processes, subset selection, and hyperparameter optimization. The idea of using GP response-sensitivity embeddings for retained-subset maintenance is practically motivated and may be useful for researchers studying large-budget BO or efficient GP surrogate updates.

**Broader Impact Concerns:**

This work presents a gradient-based sample selection method to accelerate Bayesian optimization (BO) with Gaussian process (GP) surrogates, targeting hyperparameter tuning and general optimization tasks. No prominent ethical risks, harmful biases, safety hazards, or negative societal impacts are identified from the core methodology, algorithms, and experimental evaluations of this paper.
The technique is a general-purpose computational optimization framework for machine learning and numerical optimization workflows. Its intended use includes academic research, model hyperparameter optimization, and standard industrial parameter tuning. There are no designs, experiments, or outputs that involve sensitive personal data, biometric information, surveillance systems, autonomous decision-making for high-stakes scenarios (e.g., healthcare diagnosis, autonomous vehicles, criminal justice, financial risk assessment), or discriminatory modeling.
The only real-world test case adopted is a diabetes-related hyperparameter optimization task. The experiment merely tunes model parameters for a diabetes analysis model, without conducting clinical diagnosis, treatment recommendation, or direct medical decision-making. No protected patient identifiable information is disclosed, manipulated or analyzed in the paper, and the study does not pose risks to patient privacy or medical safety.
As a efficiency-improving algorithm for general BO and sparse GP, this method may help reduce computational resource consumption for large-scale optimization workloads, which can bring mild positive impacts by cutting energy usage and hardware costs for research and industrial applications.
In summary, there are no outstanding broader impact, ethical, privacy, safety or fairness concerns that require additional dedicated Broader Impact Statement content for this submission.

**Claims And Evidence:**

No

**Claims Explanation:**

The paper provides partially supportive evidence, but the current evidence is not yet fully convincing for the main claims. The theoretical analysis in Section 5 does not directly justify the actual GSSBO selection rule. Theorem 5.1, Proposition 5.3, and Corollary 5.4 analyze posterior and UCB acquisition gaps for a fixed retained subset under shared hyperparameters. The manuscript itself acknowledges, in the remark after Corollary 5.4, that these results do not prove that the cosine-diversity selection rule controls the residual quantities appearing in the bounds. Thus, the theory is a useful diagnostic analysis of subset fitting, but it does not establish that GSSBO’s specific subset-selection rule is theoretically well motivated.
The computational-efficiency claim also requires stronger evidence. Section 4.3 states that the subset-GP refitting cost is capped at $O(M^3)$ after the switch, but the same section also states that sensitivity-embedding construction and cosine-diversity selection introduce additional costs, denoted by $C_{\mathrm{emb}}(n_t,q_t)$ and $C_{\mathrm{div}}(q_t,M)$. The paper further notes that exact full-history precision-column embeddings can require an $O(n_t^3)$ factorization, while other implementations may require $O(n_t^2)$ updates or $O(q_t^3)$ pool-specific computations. Therefore, the main scalability claim depends heavily on implementation choices. Figure 3 reports total cumulative runtime, but the paper does not decompose runtime into several parts of interests: GP refitting, embedding construction, subset selection, and other parts.
The experimental evidence is promising but incomplete. Section 6 reports 50 repeated runs with mean curves and standard-deviation bands, but the paper does not provide final cumulative-regret and runtime tables with confidence intervals or statistical tests. The ablation RSSBO is useful, but more ablations are needed to isolate the effects of the sensitivity embedding, cosine-diversity rule, latest-sample retention, dynamic buffer-size rule, and pool size. In particular, Algorithm 1 line 12 introduces a selection pool $P_t\subseteq D_t$, but the experimental section does not clearly specify how $P_t$ is constructed in all experiments.
The real-world evaluation in Section 6.2 is also limited. The diabetes hyperparameter-optimization task has only four hyperparameters, and the paper does not provide enough details about model architecture, preprocessing, train-validation split, number of epochs, stochasticity, or random seeds. This makes it difficult to assess whether the reported real-world result generalizes beyond this small example.
Finally, the dynamic buffer-size rule in Section 4.3 is hardware-dependent. The rule switches when $T_{\mathrm{current}}>Z\bar T$, and Appendix B.1 acknowledges that the switching iteration may vary across machines. This raises reproducibility concerns unless fixed-buffer or fixed-switching experiments are also reported.

**Requested Changes:**

1. Clarify the scope of the theoretical contribution and its connection to GSSBO. Section 5 and Appendix A.7 correctly state that the theoretical results are conditional diagnostics for a fixed retained subset under shared hyperparameters. However, the current contribution statement may still suggest that the theory supports the GSSBO selection rule itself. The authors should clearly state in the Introduction and contribution list that Theorem 5.1, Proposition 5.3, and Corollary 5.4 do not prove that the cosine-diversity rule controls the residual quantities in the bounds. Ideally, the authors should either add a theoretical argument or empirical diagnostics showing whether GSSBO actually reduces quantities such as $\|r_U\|$, $\|c_U(x)\|$, $\|s_R\|$, $\rho_U$, and $\|S^{-1}\|$ compared with random or alternative subset-selection rules.
2. Specify the exact subset-selection implementation, especially the construction of $P_t$. Algorithm 1 line 12 introduces a selection pool $P_t\subseteq D_t$, but the paper does not clearly specify how this pool is constructed in the experiments. This is a critical detail because both the computational cost and the quality of the retained subset depend on $q_t=|P_t|$. The authors should explicitly state whether $P_t$ is the full history, a recent window, a random candidate pool, or another prefiltered set, and should include ablations over different pool sizes.
3. Decompose the computational cost and runtime results. Section 4.3 states that the subset-GP refitting cost is capped at $O(M^3)$, but the same section also notes that embedding construction and cosine-diversity selection can introduce costs such as $C_{\mathrm{emb}}(n_t,q_t)$ and $C_{\mathrm{div}}(q_t,M)$. Since exact full-history precision-column embeddings may require an $O(n_t^3)$ factorization, the current total runtime curves in Figure 3 are not sufficient to substantiate the computational-efficiency claim. The authors should separately report the time spent on these parts.
4. Add stronger ablation studies to isolate the proposed design choices. RSSBO is a useful ablation, but it only tests random subset selection against GSSBO. The paper should further evaluate which components are actually responsible for the gains. Important ablations include sensitivity embeddings versus scalar response-gradient scores, cosine diversity versus K-means++ or max-min kernel selection, forcing the newest observation into the subset versus not forcing it, fixed buffer size versus dynamic buffer size, and different values of $M$ and $q_t$.
5. Improve the fairness and transparency of baseline comparisons. Section 6 compares GSSBO with GP-UCB, RSSBO, VecchiaBO, LR-First, SparseGP, IPA, and mini-META, but the paper does not provide enough implementation details to assess whether the comparisons are fully fair. The authors should add a table specifying, for each baseline, the acquisition function, surrogate model, subset or inducing-point size, hyperparameter learning procedure, implementation source, tuning parameters, and acquisition-optimization routine. Final cumulative regret and runtime should also be reported in tables with confidence intervals or statistical tests over the 50 runs.
6. Strengthen reproducibility and experimental documentation. The dynamic buffer rule in Section 4.3 depends on wall-clock time through $T_{\mathrm{current}}>Z\bar T$, and Appendix B.1 acknowledges that the switching iteration may vary across machines. The authors should include deterministic fixed-buffer or fixed-switching experiments in the main results. In addition, the real-world diabetes experiment in Section 6.2 should provide more details on model architecture, preprocessing, train-validation split, number of training epochs, random seeds, and whether the objective is deterministic or noisy. A clear code-release statement and scripts for reproducing all figures would substantially improve reproducibility.

---

> ### Author Response · Authors · 2026-07-17
> **Official Comment to Reviewer TGL6 Part1**
>
> We sincerely thank you for your feedback. We provide a detailed response below.
>
> > ### 1.Scope of the theoretical contribution
>
> We agree and have made the scope explicit where requested: the contribution list now states that Theorem 5.1, Proposition 5.3, and Corollary 5.4 are conditional diagnostics for *any* fixed retained subset under shared hyperparameters and do not prove that the cosine-diversity rule controls the residual quantities in the bounds.
>
> We agree that direct measurements of the residual quantities would be valuable, and the revision now reports them (Appendix C.4, Table 3). Under the paper's default retention, the vector rule attains smaller discarded-response residual $\|r_{\mathcal U}\|$, conditional cross-covariance residual $\rho_{\mathcal U}$, and realized posterior-mean gap than matched random selection on both tested tasks (by 23--38\%, geometric means of per-seed paired ratios; paired tests), while $\|s_{\mathcal R}\|$ is marginally larger (4--5\%), consistent with the selection geometry (the rule does not target this response-dependent quantity, and the near-duplicates it discards have small leave-one-out variances and hence large score entries), and no difference in $\|S^{-1}\|$ between the rules was detected, consistent with its smallest eigenvalue being governed by near-duplicate pairs that both rules discard. We are careful not to overstate this: the scalar scores and vector embeddings are algebraically related ($\mathbf s=\sum_i z_i\mathbf g_i$, $s_i=\mathbf g_i^\top\mathbf z$), and these remain empirical diagnostics on an already chosen subset. They do not close the rule-specific theoretical gap, which we continue to state explicitly.
>
> > ### 2.Exact subset-selection implementation and $P_t$
>
> We agree this was underspecified in the submission. Now stated in Section 4.3 and Algorithm 1: the selection pool always contains the forced points (the retained initial design and the latest observation) together with a discretionary candidate set, and the headline configuration takes this discretionary set to be the full history, so $\mathcal P_t=\mathcal D_t$; a restricted pool bounds only the discretionary candidates, so every forced point always carries a pool-specific embedding. The pool abstraction parameterizes the embedding and selection overheads ($C_{\mathrm{emb}}$, $C_{\mathrm{div}}$) and provides the mechanism for bounding them at larger budgets; $\mathcal P_t$ is introduced for this complexity analysis rather than as a tuning parameter, and all headline experiments use the default full-history pool. In our implementation the pool-specific embeddings are recomputed each step via one Cholesky factorization, with this cost included in the reported runtimes, and the greedy algorithm has moved into the main text. The requested pool-size and refresh ablation is added (appendix): restricting the pool does not increase regret in our ablation, and window pools further reduce the embedding overhead without increasing regret in our runs.
>
> > ### 3.Runtime decomposition.
>
> We thank the reviewer for requesting this breakdown. Added: new appendix tables (Tables 7--8) decompose total runtime into GP refitting, embedding construction, subset selection, acquisition, and a residual Other column so shares sum to 100% (up to rounding), for every method on all six benchmarks (Figures 3 and 5--6). GSSBO's total is about 9% of full-data GP-UCB's, consistent with Figure 3, and the embedding overhead is now explicit and small.
>
> > ### 4.Stronger ablations.
>
> We thank the reviewer for the concrete list. The revision adds the requested ablations (reported as mean ± 95% CI; RSSBO throughout denotes random selection with the latest sample retained): (i) vector embeddings vs scalar response-gradient scores (the vector rule attains lower regret on all six tasks); (ii) cosine diversity vs k-means++ in the same embedding space (competitive on Hart6, within noise on Eggholder2; reported as-is) plus the input-space comparison already in the appendix; (iii) forcing the newest observation on/off (minor); (iv) initial-design retention on/off, which turns out to be a large factor for the vector and random rules (its effect on the scalar rule is negligible); our reported GSSBO experiments already retain the initial design, and the revision updates Algorithm 1 to state this explicitly, and a fixed-$M$ single-factor grid with independent RNG streams isolates the selection rule from retention, confirming the vector rule is best at matched retention; (v) fixed $M\in\{50,100,200\}$ vs dynamic-$Z$ (larger $M$ moves monotonically toward full-GP performance on both tested tasks); (vi) the pool ablation of point 2. Max--min kernel selection is noted as future work. Overall takeaway: retention is a large factor for the vector and random rules (negligible for the scalar rule), and the selection-rule effect is isolated from it and reported separately.

---

> ### Author Response · Authors · 2026-07-17
> **Official Comment to Reviewer TGL6 Part2**
>
> > ### 5.Baseline fairness and statistics.
>
> We agree these details are necessary for a fair comparison. The revision adds (a) a baseline-configuration table specifying, per method, the surrogate, hyperparameter learning, implementation source, and method-specific settings under a shared acquisition and candidate-generation protocol; and (b) statistical reporting on the main results: Table 1 (mean ± 95% CI over the 50 runs) with approximate two-sample significance statements attached in the main text; GSSBO vs RSSBO is significant on four of the six tasks with no detected difference on the other two, and all outcomes are unchanged under a Holm correction. The appendix adds the runtime decomposition and CI-reported ablations. No single method dominates: the baselines that attain lower regret on individual tasks do so at a multiple of GSSBO's runtime.
>
> > ### 6.Reproducibility and documentation.
>
> We agree on all three items. (a) Deterministic fixed-$M$ results are now reported in the main results (Section 6), with the full grid and buffer tiers in the appendix; the implementation-details appendix also notes that fixing $M$ (or the switch iteration) removes the wall-clock dependence of the $Z$-trigger. (b) The diabetes task is now fully specified in Section 6.2 (dataset and split, preprocessing, architecture, optimizer, epochs, and the stochasticity of the objective) and scoped as an illustrative learning-pipeline task. (c) The code repository will be released publicly upon acceptance.

---

### Review · Reviewer_6Sag · 2026-06-30

**Summary Of Contributions:**

*Strengths*

- The problem the paper tackles, scaling Bayesian optimisation and GPs to large number of observations, is central to the application of BO and GPs in practice. This topic is of broad interest to the TMLR readership and ML community at large.

- The coverage of background material and related work in Section 1 is thorough and well-written.

- Extensive experiments are provided to demonstrate the effectiveness of the algorithm GSSBO, and the algorithm achieves strong performance in terms of regret and time complexity.

*Weaknesses*

- Some more basic material on BO would be beneficial here, the paper as written assumes familiarity with GPs and bayesian optimisation. For instance, a short section in the appendix describing the key steps in GP regression and BO (e.g. what typical choices are for the acquisition function) would make the paper more accessible to a wider audience.

- The definition of the sample embedding $g_t$ is given in Section 4.1 but it seems to be referred to already in Section 3.2. It would be useful to explicitly make a reference to the definition if it is given later in the text.

- More generally, there seems to be a lack of clarity regarding the use of the sample embedding $g_i$ and the scalar sensitivity $s_i$. For instance, the results of Prop 5.3 are formulated in terms of the scalar sensitivities but the Alg. 1 uses the vector embeddings. It would be valuable to have a more explicit discussion of what is gained/lost by using the vector embeddings instead of the scalar sensitivities, how these two are related, and if the vector embeddings $g_i$ have any direct interpretation in terms of the GP model. Right now it seems that the intuition is there for the scalar sensitivities, but the vector embeddings are used in practice due to their vector nature.

- Details on the practical implementation of subset selection are missing (see Questions below)

- Figure 2 is quite difficult to parse due to the size and the clashing plots (since many of the curves are close to each other). It would be better in my opinion to include a summary in table form.

**Audience:**

Yes

**Audience Explanation:**

The problem the paper tackles, scaling Bayesian optimisation and GPs to large number of observations, is central to the application of BO and GPs in practice and the authors provide a practical algorithm for doing this. This topic is of broad interest to the TMLR readership and ML community at large.

**Claims And Evidence:**

Yes

**Claims Explanation:**

Yes, the claim that the proposed algorithm GSSBO achieves strong performance in terms of regret and time complexity is supported by the experimental results provided in the paper.

**Requested Changes:**

*Important*
- More discussion would be needed to frame the distinction between the scalar sensitivity $s_i$ and the vector embedding $g_i$. Right now (and see my earlier comments regarding this also), it appears to me that the $s_i$ sensitivities have the interpretation of being the gradients of the GP marginal log-likelihood w.r.t. the observations, and this is the key claim in the title ("Gradient-based"). However, the embeddings $g_i$ what is being used in practice and these are columns of the precision matrix, $g_i = -K^{-1} e_i$ and do not have the direct interpretation as sensitivities/gradients of the model w.r.t. data points. The link between these two is not clear -- one immediate interpretation that lends itself is that the $g_i$ provide a kernel embedding (in terms of the negative precision matrix) of the data points $x_i$, and that the GSSBO algorithm maximises variance/diversity in this kernel embedding space.
This is, of course, totally fine - but it is important to be clear about what is actually happening in the algorithm, how it can be interpreted theoretically, and for the method to be placed in the greater context of the literature.

- Related to my point above, the theoretical results are formulated in terms of the sensitivities. These should be linked to what's done in practice, or an explanation given for why there is a gap.

- How are the samples actually selected to maximize the diversity? This seems to be addressed partially in Appendix B.1 (seems to be a greedy heuristic) but it is a key part of the algorithm, so it is important to provide this detail in the main text.

- An appendix subsection (referenced in the main paper) outlining the basics of GP regression and Bayesian optimization in the notation of the paper, geared towards readership less familiar with the field.

*Strongly encouraged*

- Summary of the results of Figure 2 in table form.

- What is the advantage of retaining an explicit subset of observations (as done in this paper) compared to using inducing points? Some more details of this in the introduction would be helpful.

---

> ### Author Response · Authors · 2026-07-17
> **Official Comment to Reviewer 6Sag**
>
> We sincerely thank you for your feedback. We provide a detailed response below.
>
> > ### 1.Background material on GPs and BO.
>
> We thank the reviewer for this suggestion. Added: an appendix Background: Gaussian Process Regression and Bayesian Optimization summarizing GP regression, the BO loop, and standard acquisition functions in the paper's notation, referenced from Section 3.1.
>
> > ### 2.Forward reference for $\mathbf g_t$.
>
> Thank you for catching this. Fixed: Section 3.2 now points explicitly to the definition in Eq. (3) of Section 4.1.
>
> > ### 3.Scalar response scores $s_i$ vs vector precision-column features $\mathbf g_i$
>
> We thank the reviewer for pressing on this distinction. Section 4.1 now states the exact relationship: $\mathbf g_i=\partial\mathbf s/\partial y_i=-\mathbf K_y^{-1}\mathbf e_i$ is a pool-specific negative precision column (the embedding view the reviewer articulated), and $s_i=\mathbf g_i^\top(\mathbf y-\boldsymbol\mu)$. The two play different roles: the response-independent embeddings supply the geometry used to *choose* a subset, while the scalar scores carry the residuals through which Proposition 5.3 *assesses* a chosen one; neither validates a selection rule, which the revision states explicitly. In a direct test under the same protocol and retention, the vector rule attains lower regret than the scalar variant on all six tasks (appendix), so it remains the default. Section 4.1 also clarifies that "gradient-based" refers to derivatives of the GP log marginal likelihood, not of the black-box objective.
>
> > ### 4.Selection implementation in the main text
>
> We agree that this detail belongs in the main text. Done: Section 4.3 now contains the greedy algorithm (initial-design points and the latest observation reserved first; incremental cumulative-cosine minimization for the remaining slots; $\mathcal O(q_tM)$ given pool scores), with the full version in the implementation-details appendix. The revision also specifies $\mathcal P_t=\mathcal D_t$ and updates Algorithm 1 to state explicitly that GSSBO retains the initial design, as in our reported experiments; an ablation quantifying its effect is reported in the appendix.
>
> > ### 5.Figure 2 as a table
>
> We thank the reviewer for the suggestion. Added: Table 1 reports the final cumulative regret of the Figure 2 experiments (mean ± 95% CI over the 50 runs), directly summarizing the figure as requested.
>
> > ### 6.Explicit subset vs inducing points
>
> We appreciate this framing suggestion. Added to the related-work discussion: the surrogate remains an exact GP on the retained data (no variational family or ELBO bias, and any acquisition function applies unmodified), the subset update is a discrete selection step rather than a continuous optimization of inducing locations, and the retained points are actual evaluated observations. The same discussion situates the precision-column embedding among gradient/influence-based data-selection heuristics and kernel-geometry sparsification, following the reviewer's framing, with a footnote anchoring the comparison empirically on Powell50.

---

### Review · Reviewer_onZK · 2026-07-04

**Summary Of Contributions:**

The paper proposes GSSBO, a subset-maintenance strategy for scaling Bayesian optimization to large-budget settings, where the O(n³) cost of GP fitting becomes prohibitive. Rather than altering the GP representation itself (as sparse/inducing-point or Vecchia-approximation methods do), GSSBO retains a fixed-size buffer of actual observed samples and refits an exact GP on that subset only. The retained subset is chosen using "sensitivity embeddings" derived from the derivative of the GP log marginal likelihood with respect to observed responses, combined with a cosine-diversity rule adapted from gradient-based sample selection in continual learning. The buffer size is set via a runtime-triggered heuristic tied to a threshold factor Z. The paper also provides conditional theoretical results such as posterior-mean/variance gap bounds, a residual-score bridge, and UCB acquisition-stability bounds for a fixed retained subset under shared hyperparameters, and reports experiments on five synthetic benchmarks and one real-world hyperparameter-tuning task against GP-UCB, RSSBO, VecchiaBO and other strong baselines, with further appendix comparisons (K-means++/Nyström,etc.)

### Strengths:

- A practically motivated problem (GP cubic scaling) tackled with a simple, interpretable mechanism (exact GP on a maintained subset, rather than an approximate posterior).
- Unusually careful theoretical scoping: the paper explicitly flags that its bounds are conditional diagnostics for a fixed subset, not proof that the selection rule minimizes the relevant residuals.
- The RSSBO ablation isolates the effect of the sensitivity-embedding/diversity rule from the effect of subsetting alone.
- Reasonably broad empirical coverage across dimensionalities, a real-world task, and several baseline families.

### Weaknesses:

- The theory does not establish why the proposed selection rule should outperform simpler alternatives (e.g., random selection); it holds for an arbitrary fixed subset.
- Empirical claims of "competitive" or "comparable" performance rest on visual comparison of mean ± 1 std over 50 runs, with no significance testing.
- Several reproducibility-relevant details are underspecified (selection pool size actually used in main experiments, hyperparameter re-estimation effects on the theory, single-machine timing).
- The buffer-size parameter Z shows inconsistent, problem-dependent effects (Fig. 4) with no guidance for practitioners on how to set it a priori.

**Additional Comments:**

The paper is generally well-written and unusually candid about the limits of its own theoretical contributions, which I view favorably. The "Scope and limitations" appendix subsection is a good practice that more papers in this area should adopt. My main concern is that the gap between the conditional theory (valid for any fixed subset) and the practical selection rule (sensitivity-embedding + cosine-diversity) is left almost entirely to the empirical section, and that section's current statistical presentation (mean ± 1 std over 50 runs, visual comparison only) is not quite strong enough to fully carry the paper's comparative claims. I would be willing to raise my assessment if the requested changes around statistical reporting and pool-size/hyperparameter clarifications are addressed.

**Audience:**

Yes

**Audience Explanation:**

The scalability of GP-based BO is a well-recognized, actively studied problem, and the idea of maintaining an explicit subset of real observations via response-sensitivity embeddings (rather than inducing points or kernel approximations) is a reasonably novel angle that connects BO to gradient-based sample-selection ideas from continual learning. Even with the limitations above, the empirical comparison against a fairly comprehensive set of scalable-BO baselines (VecchiaBO, etc. and high-dimensional embedding methods in the appendix) would be of interest to practitioners and researchers working on large-budget BO, and the conditional posterior-gap analysis (Theorem 5.1) is a clean, self-contained result that could be of independent interest for subset-of-data GP analysis more broadly, regardless of the specific selection heuristic used to produce the subset.

**Broader Impact Concerns:**

I do not see significant ethical concerns requiring a Broader Impact Statement. The work is a general-purpose optimization method with no obvious dual-use, fairness, or societal-harm considerations beyond those generic to black-box optimization and hyperparameter tuning.

**Claims And Evidence:**

No

**Claims Explanation:**

- Results are reported as mean ± 1 std over 50 runs, and comparative statements ("comparable to standard GP-UCB," "among the lowest mean cumulative regrets") are made without confidence intervals, paired statistical tests, or effect sizes. With shaded-region plots, it is difficult for a reader to assess whether the reported gaps between GSSBO, RSSBO, and GP-UCB are meaningful or within noise, particularly in Fig. 2 where several methods appear visually close.

- The theoretical analysis (Theorem 5.1, Prop 5.3, Corollary 5.4) explicitly holds kernel hyperparameters and noise variance fixed across the full-data and subset-fitted posteriors. In the actual experiments, hyperparameters are re-estimated via MLE at each iteration on the changing retained subset (Section 6). The theory does not cover the setting actually run in the experiments, and it is unclear whether the reported empirical robustness of GSSBO is attributable to the mechanism the theory describes.
- The theory doesn't support the specific design choice. The paper is honest that Theorem 5.1 and its corollaries hold for any fixed retained subset, not specifically the one selected by the cosine-diversity/sensitivity-embedding rule. This means the theoretical section, while correct, does not actually provide evidence in favor of sensitivity-embedding selection over other subset-selection heuristics (including RSSBO). The paper's justification for its central algorithmic choice is therefore almost entirely empirical, and the empirical gap between GSSBO and RSSBO is not always large (e.g., Fig. 2 on some benchmarks) or statistically substantiated.

- Algorithm 1 allows the selection pool $P_t$ to range from a restricted window to the full history ($P_t = D_t$), with materially different computational overhead ($O(q_t)$ vs. $O(n_t)$ scaling of embedding construction, per Section 4.3). The main text does not clearly state which regime is used to produce the headline runtime numbers in Fig. 3, which is important for judging whether the reported 90% runtime reduction is representative or specific to a favorable pool-size choice.

**Requested Changes:**

- Report statistical significance (e.g., paired tests or non-overlapping confidence intervals) for the main regret and runtime comparisons in Fig. 2 and Fig. 3, particularly for the GSSBO vs. RSSBO comparison, since this is the comparison that isolates the paper's core contribution.
- Explicitly state, in the main text, what selection pool $P_t$ (size and construction) is used to produce the reported runtime numbers in Section 6, and report how runtime and regret change as a function of pool size/embedding-refresh strategy (full-history vs. restricted vs. rank-one update), since this materially affects both the claimed complexity gains and reproducibility.
- Add a discussion reconciling the fixed-hyperparameter assumption in Theorem 5.1/Prop 5.3/Corollary 5.4 with the fact that hyperparameters are re-estimated via MLE at every iteration in the experiments. At minimum, discuss qualitatively (or empirically, e.g., via an ablation with fixed vs. re-estimated hyperparameters) whether this mismatch could be affecting the observed behavior.
- Provide a summary results table (final cumulative regret and final runtime, with variance) for all benchmarks and baselines, to complement the figures and support the comparative claims with concrete numbers rather than requiring readers to interpret dense multi-panel plots.

---

> ### Author Response · Authors · 2026-07-17
> **Official Comment to Reviewer onZK Part1**
>
> We sincerely thank you for your feedback. We provide a detailed response below.
>
> > ### 1.Statistical significance for the main comparisons
>
> Table 1 now reports the final values of the Figure 2 experiments (mean ± 95% CI over the 50 runs), and the main text attaches significance statements computed directly from these summary statistics (approximate two-sample tests; all stated outcomes are unchanged under a Holm correction). For the comparison the reviewer singled out, GSSBO vs RSSBO: the two default configurations share the subset-size schedule and latest-point retention but also differ in initial-design retention, so we treat Table 1 as an end-to-end comparison of the default configurations and use a fixed-$M$, matched-retention grid in the appendix to isolate the selection rule. End-to-end, GSSBO is significantly better than RSSBO on four of the six tasks and never significantly worse (no detected difference on the remaining two); at matched retention, the vector rule attains the lowest regret on both tested tasks. Versus full-data GP-UCB the outcome is task-dependent, matching Figure 2. On runtime, GSSBO's cumulative runtime at iteration 1000 is about 9% of GP-UCB's (Figure 3 and the appendix runtime table). For the GSSBO vs. RSSBO runtime comparison, Figure 3 shows similar cumulative-runtime curves, and the appendix runtime table reports final values of 211 ± 23 vs. 209 ± 22 seconds on Hart6 and 236 ± 14 vs. 218 ± 34 seconds on Powell50 (mean ± 95% CI over 50 runs). The intervals overlap, so we do not claim a significant runtime difference; Appendix Figures 5 and 6, together with Table 8, provide the corresponding runtime curves and final values (mean ± 95% CI) for the other four tasks.
>
> > ### 2.Selection pool $P_t$ and pool-size/refresh ablation.
>
> We agree that the selection-pool regime used for the headline runtime results was underspecified. The revised method specification now states $\mathcal P_t=\mathcal D_t$ (full history). The pool abstraction exists precisely to make this cost dimension explicit: the embedding and selection overheads are parameterized by the pool size ($C_{\mathrm{emb}}$, $C_{\mathrm{div}}$ in Section 4.3), and restricting the pool is the mechanism for bounding them at larger budgets. $\mathcal P_t$ is introduced for this complexity analysis rather than as a tuning parameter: practical use requires no tuning of the pool, and all headline experiments simply use the default full history. In our implementation, pool-specific embeddings are recomputed at each selection step via one Cholesky factorization, and that cost is included in the reported runtimes. Since the headline experiments use the full-history pool, the most expensive choice, with this cost included, the reported efficiency gain is not an artifact of a favorable pool restriction. The requested pool-size and refresh ablation is added (appendix): restricting the pool does not increase regret in our ablation, and window pools further reduce the embedding overhead without increasing regret in our runs, an additional optimization opportunity rather than a requirement of the reported results. Under hyperparameters frozen at the buffer switch, the cached rank-one refresh is an exact update rather than an approximation and produced identical regret to full recomputation on both tested tasks, while reducing embedding time.
>
> > ### 3.Fixed-hyperparameter theory vs per-iteration MLE.
>
> We thank the reviewer for suggesting both the reconciliation and the ablation. The revision adds a dedicated Section 5 remark reconciling the two: the theory deliberately isolates the pruning effect under shared hyperparameters, while the experiments combine it with re-estimation, so the bounds should be read as diagnostics at fixed hyperparameters. We also ran the suggested frozen-vs-re-estimated ablation (appendix): freezing moderately increases regret on both tested tasks while removing the per-iteration hyperparameter optimization, a reasonable trade-off under constrained refitting budgets; per-iteration re-estimation remains the default, and the bounds are read as diagnostics at fixed hyperparameters.
>
> > ### 4.Summary results table.
>
> We agree such a table was needed. Added: Table 1 reports final cumulative regret for all compared methods, together with the significance statements of point 1, so no comparative claim rests on visual inspection. Appendix C.5 (Runtime decomposition and component ablations) adds the baseline-configuration table (Table 6) and the runtime tables (Tables 7--8, covering all six benchmarks) with a full per-component decomposition. The takeaway: across the reported tasks and configurations, no compared method combines GSSBO's cost with better regret, and VecchiaBO's runtime exceeds full GP-UCB's.

---

> ### Author Response · Authors · 2026-07-17
> **Official Comment to Reviewer onZK Part2**
>
> > ### 5.On the noted weaknesses
>
> (i) Theory does not distinguish the rule: we agree and now state this limitation explicitly. The fixed-subset results characterize posterior and acquisition distortion after a subset has been chosen; they do not establish that vector cosine diversity, scalar-score selection, or any other heuristic controls the residual terms in the bounds. We therefore do not present the theory as evidence that the proposed rule must outperform random selection. Instead, the revision evaluates the design choice empirically through the scalar-versus-vector comparison and the fixed-$M$ rule-versus-random grid: the vector default attains lower regret than both the scalar variant and random selection on the tested tasks. A new appendix (C.4, Table 3) additionally measures the Section-5 bound quantities directly: at the paper's default retention, the vector rule yields a smaller discarded-response residual, conditional cross-covariance residual, and posterior-mean gap than matched random selection on both tested tasks. These results provide direct empirical evidence; the lack of a rule-specific theoretical guarantee remains a limitation.
>
> (ii) Guidance for $Z$: Section 6 now explains that $Z$ sets the switching threshold and hence the retained-buffer size rather than directly controlling the acquisition rule. Figure 4 shows task-dependent effects: a larger $Z$ delays switching and typically yields a larger buffer, whereas a smaller $Z$ switches earlier and limits computation more aggressively. We use $Z{=}4$ as a common reference setting, not as a universally optimal value; the fixed-$M$ experiments provide a hardware-independent alternative.